# COPI selectively drives maturation of the early Golgi

Effrosyni Papanikou[1†], Kasey J Day[1†], Jotham Austin II[2], Benjamin S Glick[1*]

[1]Department of Molecular Genetics and Cell Biology, The University of Chicago, Chicago, United States; [2]Electron Microscopy Core Facility, The University of Chicago, Chicago, United States

**Abstract** COPI coated vesicles carry material between Golgi compartments, but the role of COPI in the secretory pathway has been ambiguous. Previous studies of thermosensitive yeast COPI mutants yielded the surprising conclusion that COPI was dispensable both for the secretion of certain proteins and for Golgi cisternal maturation. To revisit these issues, we optimized the anchor-away method, which allows peripheral membrane proteins such as COPI to be sequestered rapidly by adding rapamycin. Video fluorescence microscopy revealed that COPI inactivation causes an early Golgi protein to remain in place while late Golgi proteins undergo cycles of arrival and departure. These dynamics generate partially functional hybrid Golgi structures that contain both early and late Golgi proteins, explaining how secretion can persist when COPI has been inactivated. Our findings suggest that cisternal maturation involves a COPI-dependent pathway that recycles early Golgi proteins, followed by multiple COPI-independent pathways that recycle late Golgi proteins.

*For correspondence: bsglick@uchicago.edu

†These authors contributed equally to this work

Competing interests: The authors declare that no competing interests exist.

## Introduction

The COPI coat was first visualized nearly 30 years ago on vesicles budding from Golgi cisternae (*Orci et al., 1986*). This coat was shown to consist of a soluble heptameric complex that is recruited to Golgi membranes by the small GTPase Arf1 (*Serafini et al., 1991*; *Waters et al., 1991*; *Yu et al., 2012*). During vesicle formation, COPI polymerizes to form a curved lattice that captures specific cargoes, including p24 family proteins and certain SNAREs (*Beck et al., 2009*). COPI can be divided into the B subcomplex, which consists of α-, β′- and ε-COP, and the F subcomplex, which consists of β-, δ-, γ-, and ζ-COP (*Lowe and Kreis, 1995*; *Gaynor et al., 1998*; *Lee and Goldberg, 2010*; *Jackson, 2014*). In mammalian and plant cells, COPI vesicles bud from cisternae throughout the Golgi stack except for cisternae of the *trans*-Golgi network (TGN), which produces clathrin-coated vesicles (*Ladinsky et al., 1999*; *Staehelin and Kang, 2008*; *Klumperman, 2011*). Yet despite this wealth of biochemical, morphological, and structural information, the functions of COPI have been hard to elucidate.

The strongest data implicate COPI in retrograde transport to the ER. Transmembrane ER resident proteins occasionally escape from the ER, and are then retrieved from the Golgi or ER-Golgi intermediate compartment (ERGIC) in retrograde COPI vesicles (*Szul and Sztul, 2011*; *Barlowe and Miller, 2013*). Some transmembrane ER proteins contain cytosolically-oriented C-terminal KKxx-type signals, which are recognized by COPI for retrieval to the ER (*Cosson and Letourneur, 1997*). COPI also retrieves transmembrane ER proteins that associate with the Rer1 recycling factor, as well as transmembrane ER proteins that contain arginine-based sorting signals (*Sato et al., 2001*; *Michelsen et al., 2007*). Finally, COPI plays a role in retrieving the KDEL receptor (*Orci et al., 1997*), which binds escaped luminal ER proteins. Retrograde COPI vesicles are captured at the ER

**eLife digest** Proteins play many important roles for cells, and these roles often require the proteins to be in particular locations in or around the cells. A set of cell compartments called the Golgi packages certain proteins into bubble-like structures called vesicles to enable the proteins to be used elsewhere in the cell or released to the outside of the cell, in a process called the secretory pathway. The operation of the secretory pathway requires the Golgi compartments to be continually remodeled.

Proteins and other materials can be ferried between the compartments of the Golgi by another type of vesicle. These vesicles are coated with a group, or complex, of proteins called COPI, which forms a curved lattice around the vesicles and helps them to capture the materials they will transport. However, it is not clear whether COPI is also involved in remodeling of the Golgi compartments.

Papanikou, Day et al. addressed this question using a technique called the "anchor-away method" combined with microscopy to study COPI in yeast cells. The yeast were genetically engineered so that COPI activity was effectively shut down in the presence of a drug called rapamycin. The experiments show that COPI is involved in the early stages of remodeling the Golgi compartments, but not the later stages. This finding supports the emerging view of the Golgi as a self-organizing cellular machine, and it provides a framework for uncovering the engineering principles that underlie the secretory pathway.

by the Dsl1 tethering complex in a process that involves recognition of the coat (*Ren et al., 2009*; *Zink et al., 2009*).

COPI also plays a role in intra-Golgi traffic, but the evidence is open to multiple interpretations. Initially the proposal was that COPI vesicles carry secretory cargoes forward from one Golgi cisterna to the next in a "vesicle shuttle" (*Malhotra et al., 1989*; *Orci et al., 1989*). After the discovery of COPI-mediated Golgi-to-ER recycling, the vesicle shuttle model was extended by proposing that COPI vesicles act as bidirectional carriers in the ER-Golgi system (*Pelham, 1994*; *Rothman, 1996*). However, the idea that COPI vesicles carry secretory cargoes from one cisterna to the next faced the problem that some secretory cargoes are much larger than COPI vesicles. Examples include the cell-surface scales that are secreted by certain algae, and procollagen bundles in mammalian fibroblasts (*Leblond, 1989*; *Becker et al., 1995*).

This problem was addressed by the cisternal maturation model, which states that cisternae form at the *cis* face of the Golgi, then move through the stack to the *trans* face, then finally peel off to become secretory vesicles (*Glick et al., 1997*; *Mironov et al., 1997*; *Bonfanti et al., 1998*; *Glick and Malhotra, 1998*; *Pelham, 1998*). Thus, entire cisternae could act as forward carriers for secretory cargoes. COPI vesicles have been proposed to recycle resident Golgi proteins within the organelle (*Rabouille and Klumperman, 2005*). Consistent with this idea, resident Golgi proteins have been detected in mammalian COPI vesicles (*Martínez-Menárguez et al., 2001*; *Malsam et al., 2005*; *Gilchrist et al., 2006*; *Pellett et al., 2013*; *Eckert et al., 2014*)—although conflicting results have been reported in other studies (*Orci et al., 2000a*; *Cosson et al., 2002*; *Kweon et al., 2004*)— and the localization of some yeast and plant Golgi proteins has been shown to involve COPI (*Todorow et al., 2000*; *Tu et al., 2008*; *Woo et al., 2015* ). Early versions of the cisternal maturation model postulated that COPI vesicles move in a directed fashion from older to younger cisternae (*Glick et al., 1997*; *Glick and Malhotra, 1998*). However, no mechanism for such directed movement has yet emerged, suggesting instead that COPI vesicles "percolate" bidirectionally between different cisternae (*Orci et al., 2000b*; *Day et al., 2013*). Regardless of the specific traffic pattern of COPI vesicles within the Golgi, the result is thought to be a net retrograde movement of resident Golgi proteins as the cisternae mature (*Glick and Malhotra, 1998*; *Day et al., 2013*).

The cisternal maturation model does not rule out additional roles for COPI in the traffic of secretory cargoes. For example, some secretory cargoes could move forward through the Golgi on a "fast track" involving anterograde COPI vesicles (*Pelham and Rothman, 2000*). This concept is supported by evidence that both resident Golgi proteins and secretory cargoes can be incorporated

into COPI vesicles (*Orci et al., 1997*; *Malsam et al., 2005*; *Pellett et al., 2013*). Furthermore, COPI-dependent tubules that connect heterologous cisternae have been implicated in anterograde traffic through the mammalian Golgi (*Yang et al., 2011*; *Park et al., 2015*).

A prediction of current models is that COPI should be required for secretion. According to the vesicle shuttle model, COPI carries secretory cargoes through the Golgi. According to the cisternal maturation model, COPI drives the maturation process, thereby continually regenerating the Golgi cisternae that serve as anterograde carriers for secretory cargoes.

A second prediction is that COPI should be required for Golgi maturation. Presumably, as a Golgi cisterna matures into a *trans*-Golgi network (TGN) compartment, COPI vesicles bud to remove resident Golgi proteins, thereby helping to drive the Golgi-to-TGN biochemical conversion (*Papanikou and Glick, 2014*).

Both of these predictions about the role of COPI have been tested using the yeast *Saccharomyces cerevisiae*. Golgi stacking was lost during the evolution of *S. cerevisiae* (*Mowbrey and Dacks, 2009*), but this yeast retains a compartmentalized Golgi that functionally resembles the stacked organelle seen in other organisms, with a late Golgi compartment that corresponds to the mammalian TGN (*Papanikou and Glick, 2009*; *Myers and Payne, 2013*). Surprisingly, when COPI function was disrupted in *S. cerevisiae* using thermosensitive mutant COPI subunits, secretion was reportedly inhibited for some proteins but not others (*Gaynor and Emr, 1997*). Equally surprising results were obtained when a strain with a thermosensitive mutant COPI subunit was examined by video fluorescence microscopy. Golgi maturation can be readily observed in wild-type *S. cerevisiae* cells (*Losev et al., 2006*; *Matsuura-Tokita et al., 2006*), and when the COPI mutant strain was imaged at the nonpermissive temperature, maturation was slowed but not blocked (*Matsuura-Tokita et al., 2006*). The mildness of these phenotypes has added to the uncertainty about how COPI acts in the secretory pathway.

Here, we have reexamined the functions of yeast COPI using a new approach. Yeast COPI was rapidly inactivated using the anchor-away method (*Haruki et al., 2008*; *Bharucha et al., 2013*), in which FK506-rapamycin binding protein (FKBP) was fused to an "anchor" protein while FKBP-rapamycin binding domain (FRB) was fused to a COPI subunit. Addition of rapamycin caused the FRB-tagged protein to be tethered at the anchor site, thereby preventing COPI from carrying out its cellular activities.

Fluorescence microscopy revealed that COPI inactivation generated hybrid Golgi structures that contained both early and late Golgi proteins. These structures showed unusual dynamics, with an early Golgi protein often persisting for many minutes while late Golgi proteins underwent relatively normal cycles of arrival and departure. The implication is that COPI selectively drives recycling of early but not of late Golgi proteins. After COPI inactivation, the Golgi remained partially functional, indicating that recycling of early Golgi proteins is dispensable for secretion. This analysis helps to integrate the yeast data into a broader understanding of COPI function.

## Results

### The two subcomplexes of yeast COPI localize to the early Golgi

COPI has been assumed to operate in vivo as a stable heptameric complex (*Hara-Kuge et al., 1994*; *Sahlmüller et al., 2011*; *Yip and Walz, 2011*). Although COPI can be experimentally separated into the coat-like B subcomplex and the adaptor-like F subcomplex (*Beck et al., 2009*; *Jackson, 2014*), recent structural data indicate that the two subcomplexes assemble together to generate the coat (*Dodonova et al., 2015*). However, an earlier genetic study hinted that different yeast COPI subunits might act in distinct cellular pathways (*Gabriely et al., 2007*). To examine the in vivo distribution of COPI, we used fluorescence microscopy to visualize several COPI subunits in *S. cerevisiae* (*Gaynor et al., 1998*). The tagged subunits were Sec26 (β-COP), which is part of the F subcomplex, together with either the Sec21 (γ-COP) subunit of the F subcomplex or the Ret1 (α-COP) subunit of the B subcomplex. Gene replacement was used to tag Sec26 with GFP, and then to tag Sec21 or Ret1 with mCherry. In both cases the colocalization at punctate structures was essentially complete (*Figure 1A*). This result suggests that the two subcomplexes of COPI are associated in vivo, at least when COPI is bound to membranes.

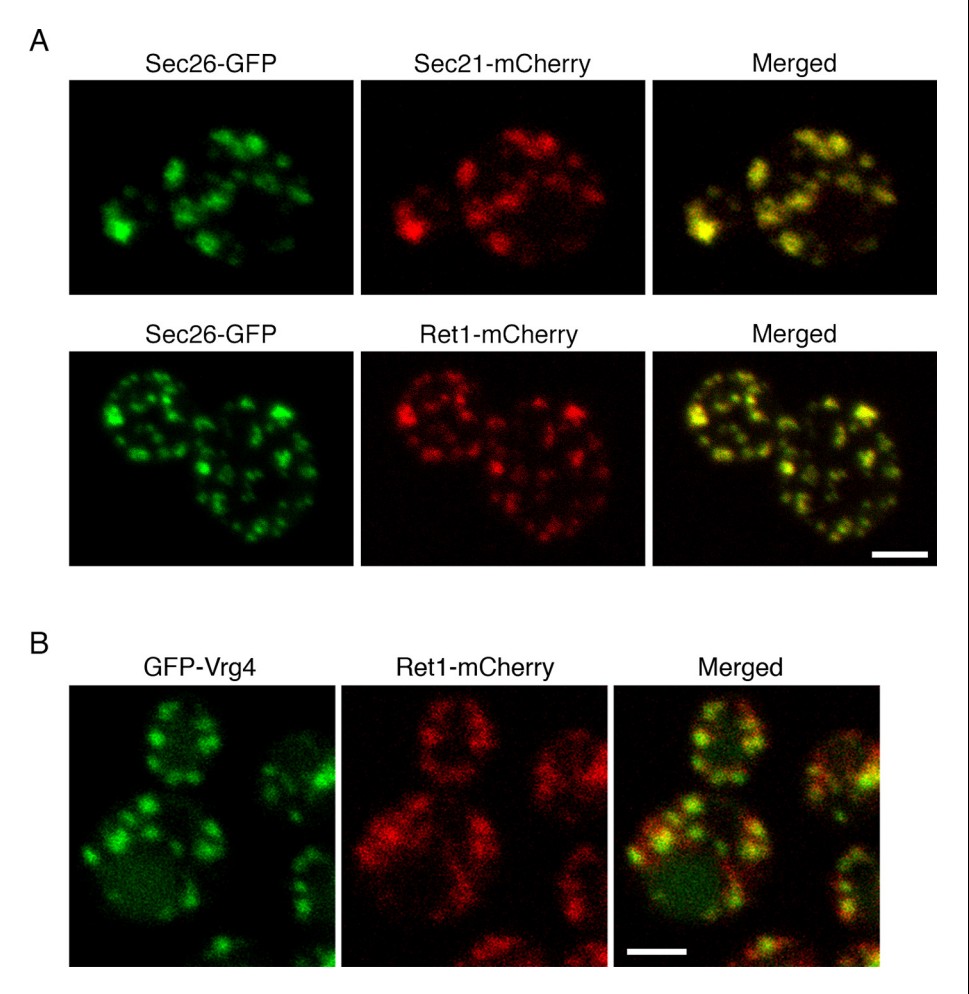

**Figure 1.** Localization of COPI subunits in yeast. (**A**) COPI subunits were tagged by gene replacement to generate Sec26-GFP, Sec21-mCherry, or Ret1-mCherry. Cells grown in NSD medium at 23°C were imaged by confocal microscopy, and two-color Z-stacks were average projected. Projections from the red and green channels were overlaid to generate merged images. Representative cells are shown. Scale bar, 2 μm. (**B**) Ret1-mCherry was expressed together with the early Golgi marker GFP-Vrg4. Imaging conditions were as in (**A**). Scale bar, 2 μm.

The COPI fluorescence pattern was somewhat variable depending on the cell being examined and the imaging conditions. Virtually all of the cells showed large COPI puncta, but a subset of the cells also showed small COPI puncta that may represent intermediates in Golgi assembly (*Figure 1A*) (*Arai et al., 2008*). COPI colocalized strongly with GFP-Vrg4, which is a GDP-mannose transporter that marks the early Golgi (*Figure 1B*) (*Dean et al., 1997*; *Abe et al., 2004*; *Losev et al., 2006*). These observations fit with electron tomography studies of mammalian and plant cells, where COPI buds were detected on Golgi but not TGN cisternae (*Ladinsky et al., 1999*; *Mogelsvang et al., 2004*; *Staehelin and Kang, 2008*). Thus, yeast COPI is likely to act at the early Golgi.

## Wild-type FRB is suitable for tagging COPI

To test the role of COPI in yeast, we turned to the anchor-away method (*Haruki et al., 2008*). The parental *S. cerevisiae* strain carried two mutations. First, the cells were rendered resistant to the growth-inhibiting effects of rapamycin by introducing the dominant *TOR1-1* point mutation (*Helliwell et al., 1994*). Second, the *FPR1* gene, which encodes the yeast homolog of FKBP, was deleted to ensure that FRB-tagged proteins would be trapped at the anchor site rather than binding

to cytosolic Fpr1 (*Lorenz and Heitman, 1995*; *Haruki et al., 2008*). The *TOR1-1 fpr1△* strain grew nearly as fast as the parental strain and was fully resistant to rapamycin.

This rapamycin-resistant strain was engineered to express the originally described ribosomal Rpl13A-FKBPx2 anchor (*Haruki et al., 2008*), and to tag yeast COPI with FRB. Some commonly used FRB variants have the T2098L mutation, which allows binding of rapamycin derivatives (*Bayle et al., 2006*). The T2098L mutation has also been reported to strengthen the affinity of FRB for FKBP-rapamycin (*Grünberg et al., 2010*). Therefore, gene replacement was used to tag Sec21, which is essential for viability (*Hosobuchi et al., 1992*), with either wild-type FRB or FRB(T2098L).

The Sec21-FRB strain grew as well as the strain carrying untagged Sec21, but the Sec21-FRB (T2098L) strain showed a pronounced growth defect (*Figure 2A*). T2098L is known to be a destabilizing mutation (*Stankunas et al., 2007*), so we suspect that FRB(T2098L) caused degradation of tagged Sec21. Interestingly, the growth defect could be rescued by appending maltose-binding protein (MPB) (*Figure 2A*) or GFP (data not shown) to the C-terminus of FRB(T2098L). After appending MBP or GFP, the destabilized FRB(T2098L) domain was no longer at an end of the polypeptide chain, and this change may have slowed degradation (*Prakash et al., 2004*; *Fishbain et al., 2011*). Regardless of the specific mechanisms at work, these results indicate that FRB(T2098L) is unsuitable as a tag for the anchor-away method, and that wild-type FRB should be used instead.

## Either ribosomes or mitochondria can serve as anchor sites

The next step was to test whether growth was inhibited when Sec21 was anchored away. Rpl13A-FKBPx2 served as an anchor on ribosomes. When this anchor was present in a rapamycin-resistant strain encoding wild-type Sec21, rapamycin addition had no effect on growth, as expected (*Figure 2B*). By contrast, when Sec21 was tagged with FRB, rapamycin completely blocked growth (*Figure 2B*). This experiment employed rapamycin at 1 µg/mL, a concentration that was sufficient for maximal growth inhibition (*Figure 2—figure supplement 1*).

A similar analysis was performed using the mitochondrial outer membrane protein OM45 (*Yaffe et al., 1989*) as an anchor. As previously observed (*Cerveny et al., 2001*), tagged and over-expressed OM45 localized to mitochondria (*Figure 2—figure supplement 2*). With a strain expressing OM45-FKBPx2 and Sec21-FRB, growth was only partially inhibited by rapamycin (*Figure 2B*). A likely explanation is that binding of FRB to FKBP-rapamycin is readily reversible (*Banaszynski et al., 2005*). If so, then why is the ribosomal anchor more effective than the mitochondrial anchor? We believe the answer is that ribosomes are present throughout the cytoplasm whereas mitochondria have a more restricted location. After dissociating from an Rpl13A-FKBPx2 anchor, Sec21-FRB will quickly be captured again by encountering another ribosome, whereas after dissociating from an OM45-FKBPx2 anchor, Sec21-FRB may have time to function at the Golgi before encountering another mitochondrion.

To test this hypothesis, we added a double FRB tag to Sec21 while increasing the number of FKBP domains on OM45 to four. These changes were predicted to increase the functional affinity of the anchoring interaction at mitochondria. Indeed, with a strain expressing OM45-FKBPx4 and Sec21-FRBx2, rapamycin completely blocked growth (*Figure 2B*). We also tried constructing a strain that expressed Rpl13A-FKBPx4 and Sec21-FRBx2, but growth was slow, perhaps because the functional affinity of the interaction was high enough to compromise COPI even in the absence of rapamycin (data not shown). In conclusion, growth assays indicate that effective anchoring can be achieved using either Sec21-FRB with a ribosomal Rpl13A-FKBPx2 anchor, or Sec21-FRBx2 with a mitochondrial OM45-FKBPx4 anchor.

As a control, we verified that anchoring of COPII also caused growth inhibition, as previously observed with *Pichia pastoris* (*Bharucha et al., 2013*). For this purpose, we added a single- or double-FRB tag to the essential COPII coat subunit Sec31 (*Salama et al., 1997*). The anchors were Rpl13A or OM45, linked to either two or four copies of FKBP. As shown in *Figure 2C*, all of the strains grew well, even the one with the Rpl13A-FKBPx4/Sec31-FRBx2 combination, and rapamycin blocked growth of all of the strains, even the one with the OM45-FKBPx2/Sec31-FRB combination. Thus, the anchor-away method is effective for inactivating components of the secretory pathway, but different components vary in their sensitivities to specific tag-anchor combinations.

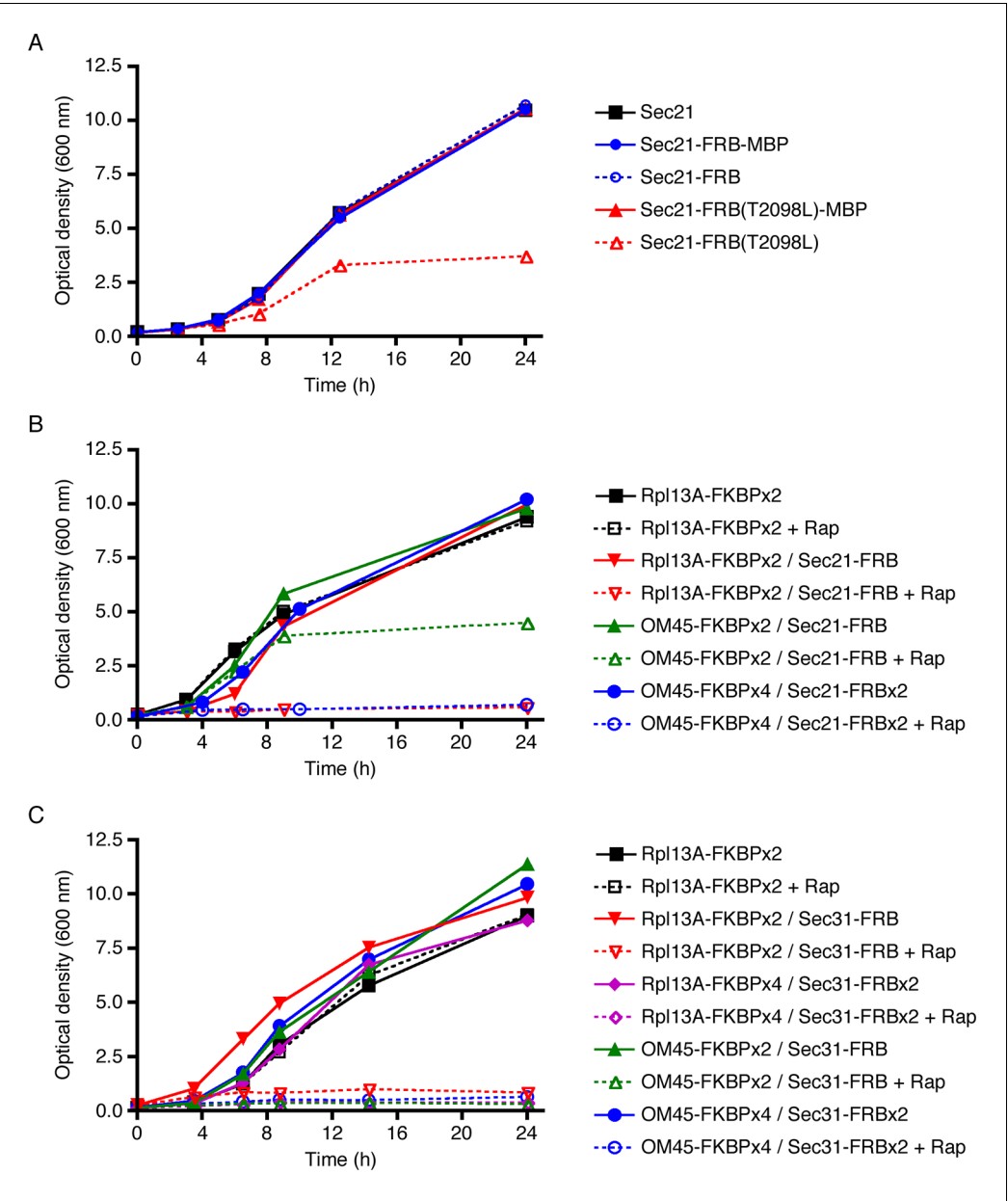

**Figure 2.** Evaluation of tag-anchor combinations by growth curve analysis. *TOR1-1 fpr1△* strains were grown with shaking in YPD medium at 30°C to mid-log phase, then diluted in fresh medium to an optical density at 600 nm (OD$_{600}$) of 0.15. Where indicated, rapamycin ("Rap") was then added to 1 μg/mL. After further incubation at 30°C, aliquots were taken at the indicated time points to measure the OD$_{600}$. (**A**) The function of Sec21 is preserved after tagging with wild-type FRB but is compromised after tagging with mutant FRB(T2098L). Gene replacement was used to extend Sec21 at the C-terminus with either wild-type FRB, or FRB(T2098L), or an FRB-MBP dual tag, or an FRB(T2098L)-MBP dual tag. A control strain expressed wild-type Sec21. (**B**) COPI can be inactivated by extending Sec21 with a single FRB tag followed by anchoring to ribosomes using the Rpl13A-FKBPx2 anchor, or by extending Sec21 with a double FRB tag followed by anchoring to mitochondria using the OM45-FKBPx4 anchor. (**C**) COPII can be inactivated by extending Sec31 with a single or double FRB tag followed by anchoring to either ribosomes or mitochondria.

The following figure supplements are available for figure 2:

**Figure supplement 1.** Effect of rapamycin concentration on anchor- mediated growth inhibition.

**Figure supplement 2.** Localization of FKBP-tagged OM45 to mitochondria.

## Anchoring Sec21 to mitochondria also displaces other COPI subunits

For further analysis, we relied mainly on the mitochondrial anchor because rapamycin-driven redistribution of labeled proteins to mitochondria can be easily visualized. The first step was to monitor the time course of the anchor-away process using Sec21-FRBx2-GFP with OM45-FKBPx4. The mitochondria were also labeled with matrix-localized mCherry. Prior to rapamycin addition, Sec21-FRBx2-GFP was present on punctate early Golgi structures that showed no consistent association with mitochondria (*Figure 3A*). At 10 min after addition of rapamycin at 23°C, Sec21-FRBx2-GFP showed substantial association with mitochondria (*Figure 3A*). Quantitation of the overlap indicated that this redistribution was maximal by 10–15 min after rapamycin addition, reaching a level of 75–80% colocalization of Sec21-FRBx2-GFP with the mitochondrial matrix marker (*Figure 3B*). We then examined strains in which Sec21-FRBx2 was nonfluorescent, with the GFP tag fused instead to other COPI subunits. Rapamycin-driven anchoring of Sec21-FRBx2 caused redistribution of Ret1-GFP or Sec26-GFP to mitochondria (*Figure 3—figure supplement 1*). Thus, the anchor-away method redistributes the entire COPI complex.

An interesting question was whether COPI became anchored to mitochondria as an isolated complex, or whether it remained bound to Golgi membranes that became tethered to mitochondria. The anchored COPI was often punctate rather than being evenly distributed over the mitochondria (*Figure 3A*), suggesting that entire Golgi compartments were being tethered. We analyzed cells that contained GFP-Vrg4 as an early Golgi marker and Ret1-mCherry as a COPI marker, plus a blue fluorescent marker for the mitochondrial matrix (*Murley et al., 2013*). Rapamycin-driven anchoring of COPI to mitochondria using Sec21-FRBx2 also displaced GFP-Vrg4 (*Figure 3C*), indicating that at least some of the anchored COPI retained its association with Golgi membranes. Although anchored COPI can still bind to the Golgi, the simultaneous anchoring to mitochondria or ribosomes evidently prevents COPI from performing its essential functions.

## Anchoring COPI inhibits recycling to the ER

If anchored COPI is functionally inactive as suggested by the growth curve analysis, then recycling of proteins from the Golgi to the ER should be inhibited. To test this prediction, we used a construct encoding the transmembrane domain (TMD) of the ER protein Sec71 fused to GFP (*Sato et al., 2003*). Sec71TMD-GFP localizes to the ER with the aid of the retrieval receptor Rer1 (*Sato et al., 2003*), which recycles from the Golgi to the ER in a COPI-dependent manner (*Boehm et al., 1997*; *Sato et al., 2001*). Control experiments confirmed that Sec71TMD-GFP was in the ER as indicated by nuclear envelope fluorescence plus additional fluorescence from peripheral ER structures (*Sato et al., 2003*). When Sec71TMD-GFP was expressed using an integrating vector in a wild-type strain, a bright ER signal was seen in essentially all of the cells (*Figure 4A*). By contrast, in an *rer1△* mutant strain, the Sec71TMD-GFP signal in the ER was faint (*Figure 4A*). The *rer1△* strain showed a vacuolar lumen signal as previously reported (*Sato et al., 2003*), although the strength of this signal varied between cells, probably because older cells had degraded more mislocalized Sec71TMD-GFP molecules and had therefore accumulated more GFP in the vacuole. These results validate the use of ER localization of Sec71TMD-GFP as a readout for Golgi-to-ER recycling.

We predicted that inactivation of COPI would prevent Rer1-dependent Golgi-to-ER recycling while allowing ER export, and would therefore reduce Sec71TMD-GFP levels in the ER. This reduction was expected to be gradual because Sec71TMD-GFP lacks a known ER export signal and should be incorporated at a low efficiency into COPII vesicles (*Dancourt and Barlowe, 2010*). In a control strain that contained the mitochondrial OM45-FKBPx4 anchor, addition of rapamycin had no effect on the amount of Sec71TMD-GFP in the ER, as indicated by the fluorescence signal from the nuclear envelope (*Figure 4B,C* and *Figure 4—figure supplement 1*). By contrast, when the OM45-FKBPx4 strain also contained Sec21-FRBx2 for anchoring COPI, addition of rapamycin caused progressive depletion of Sec71TMD-GFP from the ER (*Figure 4B,C* and *Figure 4—figure supplement 1*). After 60 min of rapamycin treatment, the level of Sec71TMD-GFP in the ER was comparable to that seen in the *rer1△* strain (dashed line in *Figure 4C*). This observation provides further evidence that anchoring to mitochondria inactivates COPI.

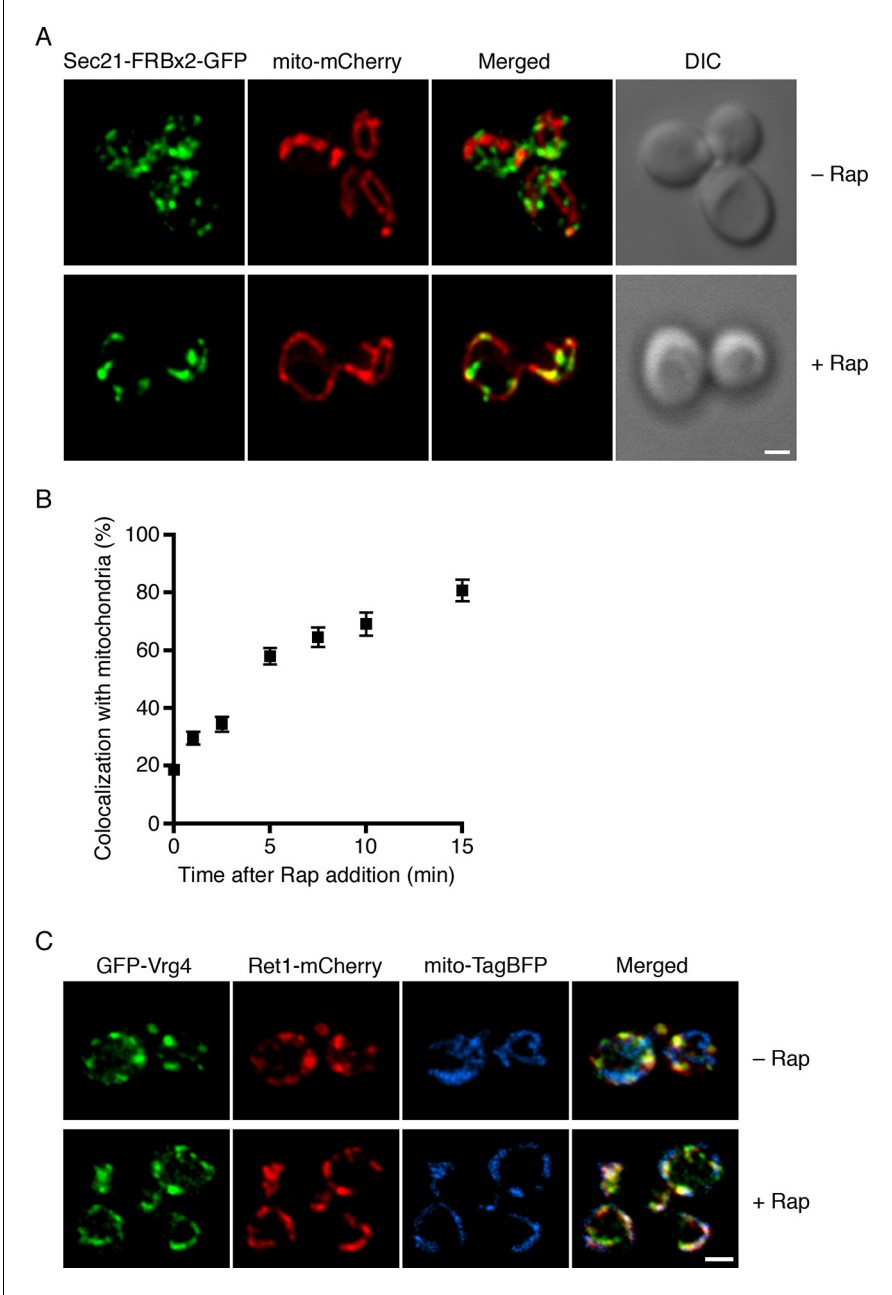

**Figure 3.** Visualization of COPI anchoring to mitochondria. (**A**) Sec21 tagged with FRBx2-GFP can be anchored to mitochondria containing OM45-FKBPx4. The mitochondrial matrix was labeled with mCherry. Representative cells are shown before drug treatment, and after treatment for 10 min at 23°C with 1 μg/mL rapamycin ("Rap"). Cells were imaged by confocal microscopy followed by deconvolution. Scale bar, 2 μm. (**B**) Maximal anchoring to mitochondria occurs within 10–15 min. Anchoring of Sec21-FRBx2-GFP was quantified by measuring the fraction of the GFP signal that colocalized with the mCherry signal (***Levi et al., 2010***) at different times after rapamycin addition at 23°C. Approximately 20–30 cells were analyzed at each time point. Error bars show s.e.m. (**C**) Anchoring COPI also anchors early Golgi cisternae. Sec21 was tagged with FRBx2 in a strain with the OM45-FKBPx4 mitochondrial anchor. The strain also expressed GFP-Vrg4 to label early Golgi cisternae, Ret1-mCherry to label COPI, and mito-TagBFP to label the mitochondrial matrix. Cells were imaged before or after rapamycin addition as in (**A**).

The following figure supplement is available for figure 3:

**Figure supplement 1.** Simultaneous anchoring of multiple COPI subunits to mitochondria.

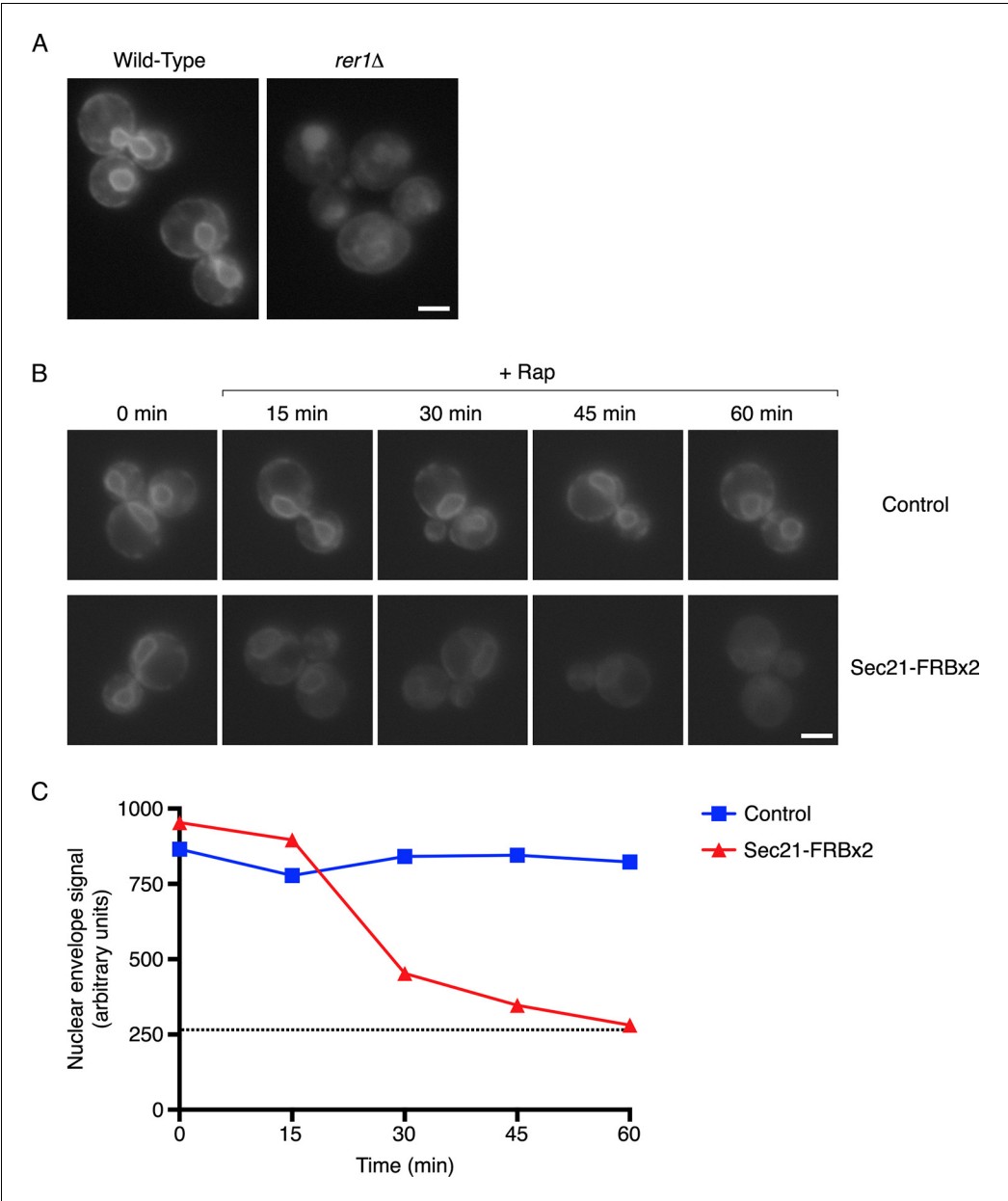

**Figure 4.** Depletion of Sec71TMD-GFP from the ER after COPI inactivation. (**A**) Sec71TMD-GFP requires Rer1 for normal ER localization. An expression vector for Sec71TMD-GFP was integrated into the wild-type JK9-3da strain or into an isogenic *rer1Δ* strain. Cells grown to log phase in NSD at 30°C were compressed beneath a coverslip and visualized in a single focal plane by widefield microscopy. Representative images taken at the same exposure are shown. Scale bar, 2 μm. (**B**) Anchoring of COPI causes loss of Sec71TMD-GFP from the ER. Both of the strains shown here contained integrating vectors for expressing the mitochondrial anchor OM45-FKBPx4 as well as Sec71TMD-GFP, and one strain also expressed Sec21-FRBx2 to anchor COPI. Cells growing at 30°C were incubated for 10 min with 0.0002% Hoechst 33342 to stain nuclei, followed by an additional treatment for up to 60 min with 1 μg/mL rapamycin ("Rap"). At the indicated time points after rapamycin addition, cells were compressed beneath a coverslip and visualized in a single focal plane by widefield microscopy. Representative images of GFP fluorescence are shown. Scale bar, 2 μm. (**C**) The data from (**B**) were quantified by measuring the GFP fluorescence signals from nuclei as a proxy for the ER signals. For this purpose, the signal in the blue channel was used to select nuclei in ImageJ using the Magic Wand tool. These selections were expanded by 12 pixels to include the nuclear envelope, and the GFP fluorescence from the selected areas in the green channel was quantified. Plotted are average fluorescence signals per unit area. Each data point was derived from approximately 40–80 cells, and was adjusted by subtracting an average background signal from nuclei in the parental strain lacking GFP. The dashed line represents the average background-corrected nuclear fluorescence signal from *rer1Δ* cells that exhibited low to moderate vacuolar fluorescence.

The following figure supplement is available for figure 4:

*Figure 4 continued on next page*

*Figure 4 continued*

**Figure supplement 1.** Labeling of nuclei to quantify Sec71TMD-GFP fluorescence in the nuclear envelope.

## Anchoring COPI reduces but does not block secretion

To determine how COPI inactivation affects forward traffic through the secretory pathway, we examined general secretion. Cells were incubated at 30°C with $^{35}$S-labeled amino acids during a 10-min pulse and then chased for 30 min, followed by analysis of the radiolabeled proteins secreted into the culture medium (*Gaynor and Emr, 1997*). This procedure was performed either with untreated cells, or with cells that had received rapamycin 10 min before the pulse.

In a control strain that contained the mitochondrial OM45-FKBPx4 anchor, a series of radioactive protein bands was detected in the absence of rapamycin, and this pattern was unchanged by addition of rapamycin (*Figure 5A*). However, when the OM45-FKBPx4 strain also contained Sec31-FRBx2 for anchoring the outer layer of the COPII coat, general secretion was completely blocked by rapamycin addition (*Figure 5A*). As a control, rapamycin had no effect on cellular protein synthesis (*Figure 5—figure supplement 1*). The inhibitory effect of anchoring Sec31-FRBx2 was expected because proteins following the conventional secretory pathway all require COPII to leave the ER (*Barlowe and Miller, 2013*).

When the OM45-FKBPx4 strain also contained Sec21-FRBx2 for anchoring COPI, general secretion was reduced but not abolished (*Figure 5A*). We extended this test by comparing three different anchoring configurations for COPI (*Figure 5B*). In a strain that contained the ribosomal Rpl13A-FKBPx2 anchor and Sec21-FRB, rapamycin reduced secretion to a similar extent as in the strain that contained the mitochondrial OM45-FKBPx4 anchor and Sec21-FRBx2. However, in a strain that contained the mitochondrial OM45-FKBPx2 anchor and Sec21-FRB, rapamycin had only a marginal effect on secretion. These results fit with the growth tests indicating that a single copy of FRB is sufficient to inactivate COPI using the ribosomal anchor, but that two copies of FRB are needed using the mitochondrial anchor (see above).

*Figure 5A* shows that after anchoring COPI, the secretion of some proteins was only weakly inhibited as marked by the arrow, whereas the secretion of other proteins was strongly inhibited as marked by the arrowhead. Many proteins fell between these two extremes, showing extensive but not complete inhibition. Similar results were obtained when the rapamycin concentration was doubled to 2 μg/mL and the drug pretreatment was varied between 5 and 30 min (*Figure 5—figure supplement 2*). Moreover, when the FRBx2 tag was placed on Ret1 instead of Sec21, rapamycin inhibited growth and anchored COPI but only partially inhibited secretion (*Figure 5—figure supplement 3*), mimicking the effects seen with Sec21-FRBx2. Even when both Sec21 and Ret1 were tagged with FRBx2, rapamycin did not fully inhibit secretion (*Figure 5—figure supplement 3*). Based on these data, we conclude that COPI inactivation allows general secretion to continue at a reduced level, with the magnitude of the reduction varying for different proteins.

## The effects of anchoring COPI resemble the effects of the *sec21-3* mutation

Because the effects of anchoring COPI are similar to those previously seen after thermal inactivation of COPI in a *sec21-3* mutant (*Gaynor and Emr, 1997*), we performed a direct comparison of the two approaches. Sequencing of genomic DNA from a *sec21-3* mutant strain revealed that this allele carries a single F720S point mutation. We introduced this mutation into our parental strain by gene replacement. As anticipated, the *sec21-3* mutant cells grew at 23°C but not at 37°C (*Figure 6—figure supplement 1*).

The *sec21-3* mutant was shifted to 37°C for 30 min prior to pulse-chase analysis. In parallel, anchoring to mitochondria was performed at 37°C. As shown in *Figure 6*, the *sec21-3* mutation had the same effect on general secretion as anchoring Sec21-FRBx2. In both cases, secretion was inhibited to varying extents for different proteins. Anchoring Sec31-FRBx2 at 37°C completely blocked general secretion (*Figure 6*), confirming that secretion has a more stringent requirement for COPII than for COPI.

These results suggested that the *sec21-3* mutant strain might be useful for a fluorescence microscopy analysis of how COPI inactivation affects the Golgi. Unfortunately, when the *sec21-3* strain was

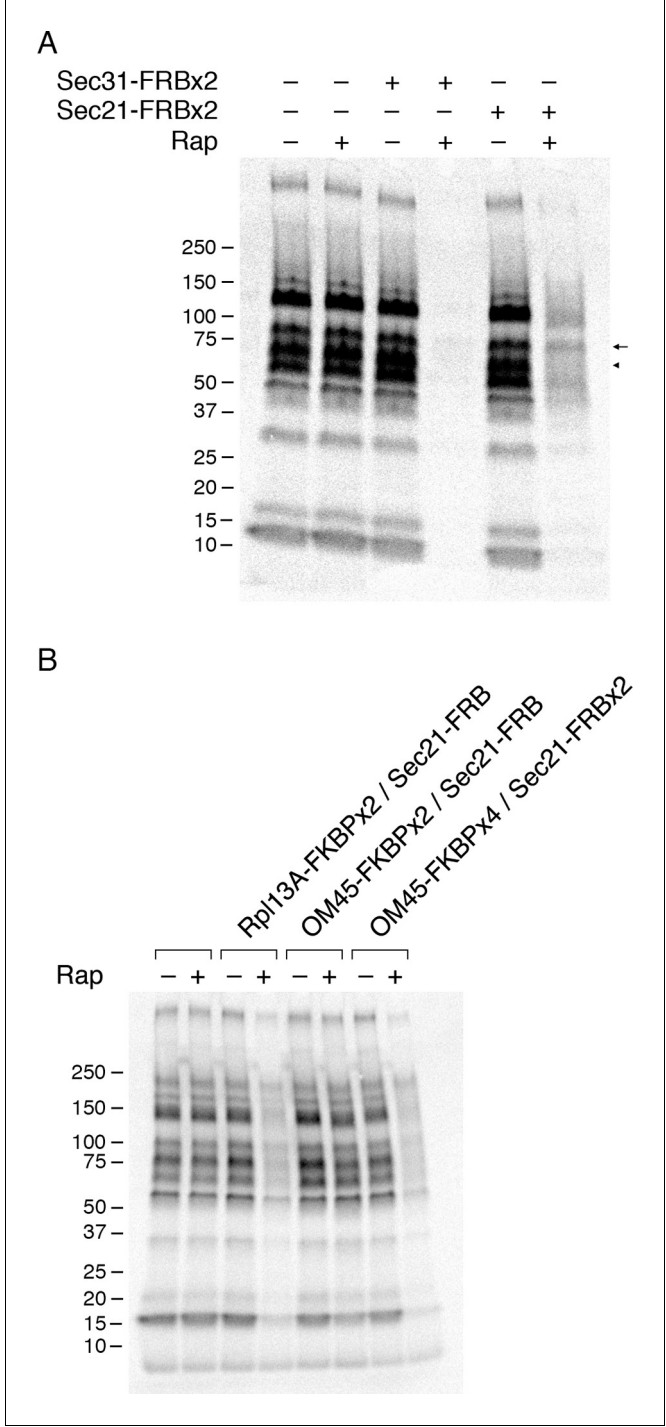

**Figure 5.** Effects of anchoring COPI or COPII on general secretion. (**A**) Anchoring COPII inhibits secretion more strongly than anchoring COPI. The yeast strains used here all expressed the mitochondrial anchor OM45-FKBPx4. Where indicated, a strain also expressed either Sec31-FRBx2 to anchor COPII, or Sec21-FRBx2 to anchor COPI. Cells growing at 30°C were either left untreated or treated for 10 min with 1 μg/mL rapamycin ("Rap"), then pulsed for 10 min with $^{35}$S amino acids, then chased for 30 min. The culture medium was analyzed by SDS-PAGE and autoradiography to detect secreted proteins that had been labeled during the pulse. Numbers represent the molecular weight in kDa of marker proteins. After anchoring COPI, the arrow marks a band of secreted protein that was only partially diminished, and the arrowhead marks a band of secreted protein that was severely diminished. (**B**) Efficient anchoring of Sec21 requires two copies of FRB with a mitochondrial anchor but only one copy of FRB with a ribosomal anchor. General secretion was visualized by a pulse-chase procedure as in (**A**),

*Figure 5 continued on next page*

Papanikou *et al*. eLife 2015;4:e13232. DOI: 10.7554/eLife.13232

*Figure 5 continued*

except that the control strain expressed neither an anchor nor an FRB-tagged protein. Where indicated, a strain expressed either a ribosomal Rpl13A-FKBPx2 anchor or a mitochondrial OM45-FKBPx2 or OM45-FKBPx4 anchor, plus either Sec21-FRB or Sec21-FRBx2.

The following figure supplements are available for figure 5:

**Figure supplement 1.** Insensitivity of cellular protein synthesis to rapamycin.

**Figure supplement 2.** Time course of the rapamycin effect on general secretion.

**Figure supplement 3.** Partial inhibition of secretion after anchoring different COPI subunits.

---

transformed to express the early Golgi marker GFP-Vrg4 and the late Golgi marker Sec7-DsRed (*Losev et al., 2006*), Golgi organization was perturbed even at the permissive temperature. Compared to a wild-type control strain, the size of the labeled puncta was more variable, and a greater fraction of the GFP-Vrg4 marker was found in non-punctate structures (*Figure 6—figure supplement 2*). Thus, the anchor-away approach is preferable for microscopy because Golgi organization should be normal prior to rapamycin addition.

## Anchoring COPI leads to formation of hybrid Golgi structures

We asked whether Golgi organization was altered by inactivating COPI. The first test employed fluorescence microscopy with GFP-Vrg4 and Sec7-DsRed. As previously observed (*Losev et al., 2006*), early and late cisternae appeared as distinct puncta in the absence of rapamycin (*Figure 7A*). In a strain carrying the OM45-FKBPx4 anchor and wild-type Sec21, addition of rapamycin had no effect on this pattern (*Figure 7A*). However, in a strain carrying OM45-FKBPx4 plus Sec21-FRBx2, addition

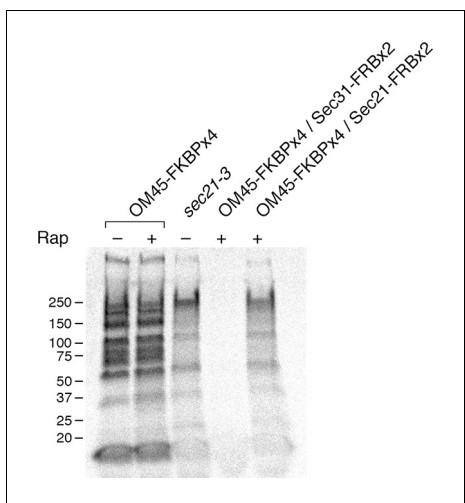

**Figure 6.** Comparison of methods for inactivating Sec21. General secretion was visualized by a pulse-chase analysis as in *Figure 5A*, except that cells were grown at 23°C and then shifted to 37°C for 30 min before the procedure. A control strain expressed the mitochondrial OM45-FKBPx4 anchor. Where indicated, a strain expressed OM45-FKBPx4 plus either Sec31-FRBx2 to anchor COPII or Sec21-FRBx2 to anchor COPI. The *sec21-3* mutation was in the parental strain background. "Rap" indicates a 10-min treatment with 1 μg/mL rapamycin prior to the pulse. Numbers represent the molecular weight in kDa of marker proteins.

The following figure supplements are available for figure 6:

**Figure supplement 1.** Thermosensitivity of the *sec21-3* strain.

**Figure supplement 2.** Perturbed Golgi structure in *sec21-3* cells at the permissive temperature.

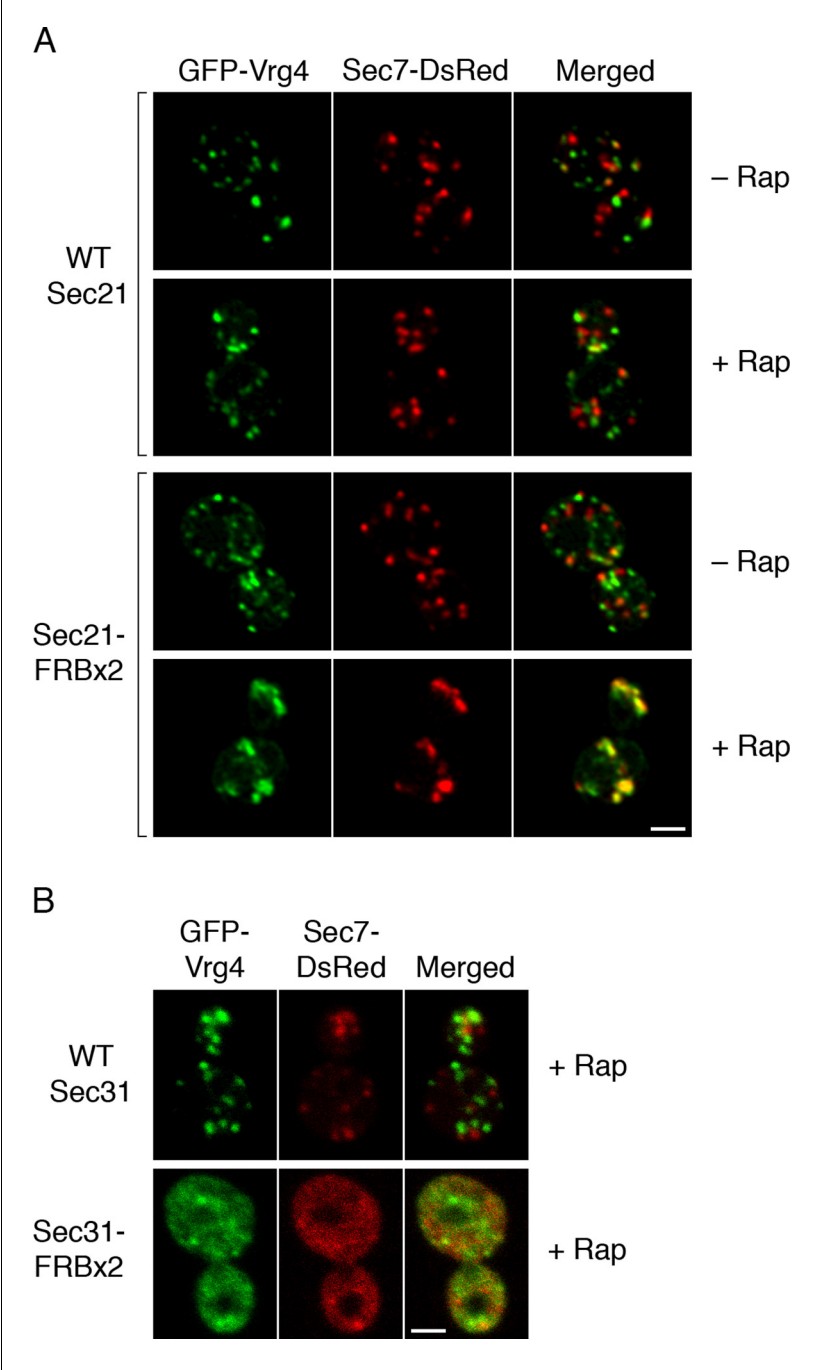

**Figure 7.** Effects of anchoring COPI or COPII on Golgi organization. (A) Anchoring COPI leads to association of early and late Golgi markers. A strain carrying the OM45-FKBPx4 anchor was transformed to express the early Golgi marker GFP-Vrg4 and the late Golgi marker Sec7-DsRed. A derivative strain also expressed Sec21-FRBx2 to anchor COPI. Cells were grown and imaged as in *Figure 1*, except that the confocal images were deconvolved. "+ Rap" indicates a 10-min treatment with 1 μg/mL rapamycin prior to imaging. Scale bar, 2 μm. (B) Anchoring COPII does not lead to association of early and late Golgi markers. The experiment was performed with rapamycin addition as in (A), except that the strain expressed Sec31-FRBx2 to anchor COPII, and deconvolution was omitted. Similar results were seen when COPII was anchored to mitochondria by incubating with rapamycin for 10 min as in the figure, or for 20 min (data not shown).

The following figure supplements are available for figure 7:

**Figure supplement 1.** Formation of hybrid Golgi structures with a ribosomal anchor.

*Figure 7 continued on next page*

*Figure 7 continued*

**Figure supplement 2.** Visualization of hybrid Golgi structures with an alternative early Golgi marker.

of rapamycin for 10 min to anchor COPI caused a dramatic change: the labeled structures were fewer in number and often larger, and many of them showed colocalization of the early and late Golgi markers (*Figure 7A*). The green and red fluorescence patterns were typically not identical, suggesting that the early and late Golgi markers were associated but partially segregated. This altered distribution of Golgi markers remained largely unchanged after prolonged treatment with rapamycin, except that after 30 min, some of the GFP-Vrg4 began to appear at the vacuolar membrane and plasma membrane (data not shown). A similar conversion of separate to partially overlapping localizations of early and late Golgi markers was seen when a ribosomal anchor was used instead of a mitochondrial anchor (*Figure 7—figure supplement 1*), or when the early Golgi marker was the Cog1 subunit of the COG vesicle tethering complex (*Figure 7—figure supplement 2*) (*Willett et al., 2013*). We infer that COPI inactivation generates hybrid Golgi structures containing both early and late Golgi markers.

A concern was that formation of hybrid Golgi structures might be a nonspecific consequence of perturbing membrane traffic. To exclude this possibility, we visualized Golgi markers after

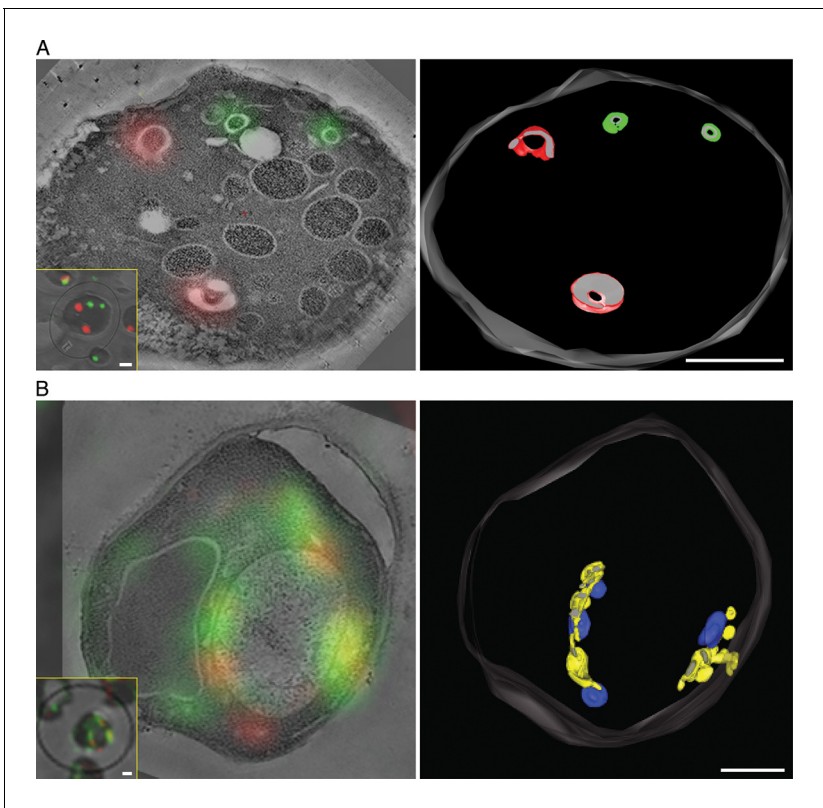

**Figure 8.** Correlative fluorescence microscopy and electron tomography of the yeast Golgi. The yeast strain expressed GFP-Vrg4 and Sec7-DsRed, as well as OM45-FKBPx4 as a mitochondrial anchor plus Sec21-FRBx2 to anchor COPI. Cells were frozen, freeze substituted, and embedded in plastic in a manner that preserved fluorescence. Plastic sections were then examined by fluorescence microscopy followed by STEM tomography. The figures show either (**A**) a section ~0.5 μm thick from a representative untreated cell, or (**B**) a section ~1.0 μm thick from a representative cell after treatment for 10 min with 1 μg/mL rapamycin. In each case, the left panel shows the fluorescence signals overlaid on a projection of five 3- to 4-nm tomographic slices, the inset shows fluorescence and differential interference contrast analysis of the same cell, and the right panel shows a tomographic model of the relevant labeled structures. Full tomographic reconstructions for the untreated and rapamycin-treated cells are animated in *Video 1* and *Video 2*, respectively. Green indicates early Golgi cisternae, red indicates late Golgi cisternae, yellow indicates hybrid Golgi structures, and blue indicates mitochondria. Scale bars, 1 μm.

inactivating COPII. When a strain carrying OM45-FKBPx4 plus Sec31-FRBx2 was treated with rapamycin to anchor COPII, most of the GFP-Vrg4 and Sec7-DsRed were dispersed in the cytoplasm, with the remaining punctate structures showing little colocalization of the two markers (*Figure 7B*). This result is consistent with an earlier report that inactivation of yeast COPII disrupts the Golgi (*Morin-Ganet et al., 2000*). Thus, the association of early and late Golgi markers after anchoring COPI is a specific effect.

To visualize the hybrid Golgi structures at higher resolution, we built on our previous experience with imaging GFP-tagged yeast compartments by correlative fluorescence microscopy and electron tomography (*Bhave et al., 2014*). This method was extended to dual-color imaging by visualizing both GFP-Vrg4 and Sec7-DsRed in plastic-embedded samples (*Figure 8*, insets), and was further enhanced by examining thicker sections with scanning transmission electron microscopy (STEM) tomography (*Aoyama et al., 2008*; *Hohmann-Marriott et al., 2009*; *Sousa et al., 2011*). *Figure 8A* shows Golgi cisternae in untreated cells. Individual early and late Golgi cisternae are identifiable in the tomograms by correlation with the fluorescence data. The tomographic reconstruction (*Video 1*) indicates that the Golgi cisternae were relatively simple curved structures (*Preuss et al., 1992*; *Bhave et al., 2014*), and that the early and late cisternae were not visibly connected. *Figure 8B* shows Golgi structures after COPI inactivation. The tomographic reconstruction (*Video 2*) indicates that the early and late Golgi markers were present in irregularly shaped membrane compartments of various sizes. As predicted from the fluorescence analysis (see above), some of the Golgi membranes were closely apposed to mitochondria. We conclude that COPI inactivation converts the yeast Golgi into complex membrane structures that can contain both early and late Golgi proteins.

## Anchoring COPI selectively blocks recycling of an early Golgi marker

How does COPI inactivation generate hybrid Golgi structures? This question was addressed using 4D confocal microscopy. We previously showed that in unperturbed cells expressing GFP-Vrg4 and Sec7-DsRed, a cisterna labels for ~1–3 min with the green early Golgi marker, then loses green fluorescence while acquiring red fluorescence, then labels for ~1–3 min with the red late Golgi marker, then loses red fluorescence (*Losev et al., 2006*). The same maturation dynamics for GFP-Vrg4 and Sec7-DsRed were seen when a strain expressing the OM45-FKBPx4 anchor plus Sec21-FRBx2 was imaged in the absence of rapamycin (data not shown). By contrast, after the same strain had been incubated with rapamycin for 10 min to inactivate COPI, Golgi dynamics were markedly altered as described below. Rapamycin-driven tethering to mitochondria probably facilitated this analysis by making the Golgi structures less mobile and easier to track.

After COPI inactivation, two behaviors were evident from the 4D movies (*Video 3*). First, when the early and late Golgi markers were both present in a hybrid structure, the two markers remained closely associated even

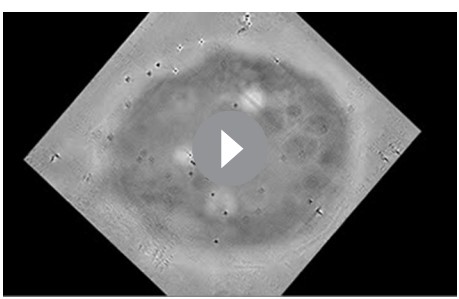

**Video 1.** Tomographic reconstruction of Golgi cisternae in an untreated cell. Yeast cells were analyzed by STEM tomography as described in *Figure 8*. The first third of the movie shows every tenth tomographic slice. The second third of the movie shows the same tomographic slices after contours were traced to label early Golgi membranes green and late Golgi membranes red. The final third of the movie shows a rotation of the tomographic model. See also *Figure 8A*.

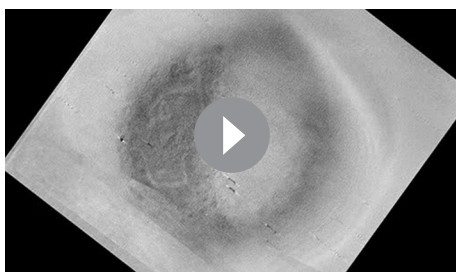

**Video 2.** Tomographic reconstruction of hybrid Golgi structures in a cell that was treated with rapamycin to inactivate COPI. The procedure was the same as for *Video 1*, except that COPI was inactivated prior to freezing the cells. Putative Golgi membranes are labeled yellow and mitochondrial outer membranes are labeled blue. See also *Figure 8B*.

though the shape of the structure fluctuated rapidly. Second, in many of the hybrid Golgi structures, the Sec7-DsRed marker disappeared and then subsequently returned while the GFP-Vrg4 marker persisted the entire time. An example from *Video 3* is the Golgi structure marked by an arrowhead in *Figure 9—figure supplement 1*. The dynamics of this structure are illustrated in *Video 4*, which was generated by cropping the movie frames and then manually erasing the fluorescence of nearby structures at each time point. *Figure 9A* shows representative frames from *Video 4* together with a quantitation of the green and red fluorescence intensities for the entire 15-min movie. The red signal lasted for ~4.5 min, then completely disappeared, then returned for ~7 min, then briefly disappeared before beginning to increase once again. A similar pattern was seen with the structure marked by the arrow in *Figure 9—figure supplement 1*, although this structure could be tracked reliably for only ~8.5 min, as illustrated in *Video 5*. Representative frames from *Video 5* are shown in *Figure 9B*. In this case, the red signal disappeared for ~2 min before returning even stronger than before. More generally, although the kinetics varied for different Golgi structures, many of them showed persistent GFP-Vrg4 labeling with alternating high and low levels of Sec7-DsRed labeling. Thus, Sec7-DsRed continued to recycle within the Golgi after COPI inactivation.

## Anchoring COPI Does Not Block Recycling of a Late Golgi Transmembrane Protein

Because Sec7 is a late Golgi peripheral membrane protein, we wondered whether late Golgi transmembrane proteins would also continue to recycle after COPI inactivation. A late Golgi transmembrane protein that may recycle within the Golgi is the processing protease Kex2 (*Fuller et al., 1988*). It has been proposed that Kex2 traffics between the Golgi and prevacuolar endosomes (*Sipos et al., 2004*; *De et al., 2013*), but Kex2 has no known role at prevacuolar endosomes, and the following two results suggest that the major recycling pathway for Kex2 is actually within the Golgi.

First, we looked for colocalization of Kex2 with markers of the late Golgi or prevacuolar endosomes. The marker for the late Golgi was Sec7. The marker for prevacuolar endosomes was Vps8, a subunit of the CORVET tether (*Arlt et al., 2015*). Late Golgi compartments and prevacuolar endosomes are clearly distinct because in projected confocal images, Sec7-mCherry and Vps8-GFP showed no colocalization apart from a background level that likely represents chance overlap after projection (*Figure 10*) (*Arlt et al., 2015*). As a control, we examined Vps10, a vacuolar

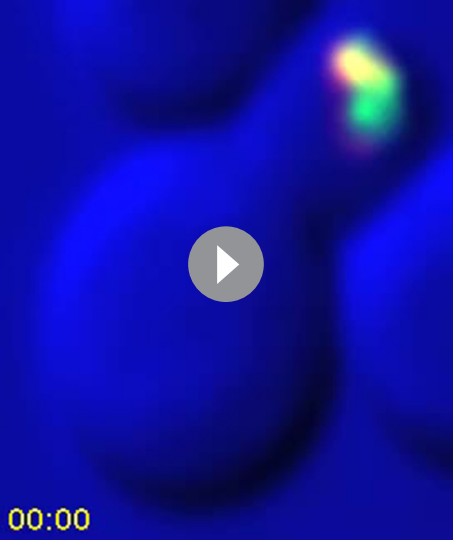

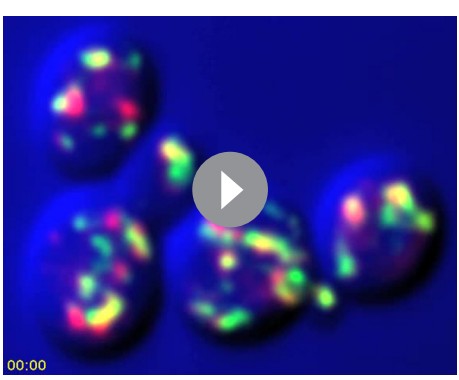

**Video 3.** Dynamics of Vrg4 and Sec7 after anchoring COPI. Cells in which Sec21 had been anchored to mitochondria were imaged by dual-color 4D confocal microscopy to visualize the dynamics of the early Golgi marker GFP-Vrg4 and the late Golgi marker Sec7-DsRed. Scattered light images were recorded in the blue channel. Complete Z-stacks were collected every 2 s for 15 min, and the data were deconvolved, bleach corrected, and average projected. See also *Figure 9*.

**Video 4.** Edited movie showing Golgi Structure 1 from *Video 3*. *Video 3* was cropped to include only a single budded cell. At each time point, a montage of the Z-stack was generated, and fluorescence signals were erased for all structures except the one marked with the arrowhead in *Figure 9—figure supplement 1*. This structure was designated Golgi Structure 1. An edited version of the cropped movie was then generated. See also *Figure 9A*.

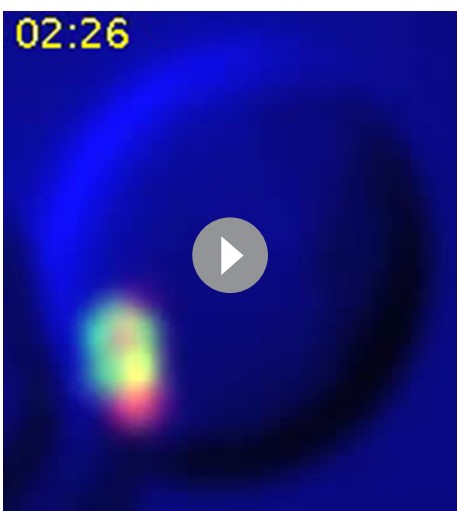

**Video 5.** Edited movie showing Golgi Structure 2 from *Video 3*. *Video 3* was cropped to include only a single cell. The procedure was the same as for *Video 4*, except that this edited movie displays Golgi Structure 2, which is marked with an arrow in *Figure 9—figure supplement 1*. See also *Figure 9B*.

hydrolase receptor that cycles between the late Golgi and prevacuolar endosomes and localizes to both compartments (*Marcusson et al., 1994*; *Cooper and Stevens, 1996*; *Chi et al., 2014*). Vps10-GFP was present in a subset of the late Golgi structures labeled with Sec7-mCherry and in almost all of the prevacuolar endosomes labeled with Vps8-mCherry (*Figure 10*). By contrast, Kex2-GFP colocalized strongly with Sec7-mCherry but showed very little colocalization with Vps8-mCherry (*Figure 10*). In some cells, prevacuolar endosomes did contain detectable Kex2-GFP, but this green signal was weak (*Figure 10—figure supplement 1*). These results suggest that Kex2 resides in the Golgi, and that

cycling through prevacuolar endosomes represents at most a minor pathway for Kex2 trafficking.

Second, we used 4D confocal microscopy to compare the dynamics of Kex2-GFP and Sec7-mCherry, and to determine why the colocalization of these two markers is substantial but incomplete (*Franzusoff et al., 1991*; *Redding et al., 1991*). It was recently reported that Kex2 departs from Golgi cisternae somewhat earlier than Sec7 (*McDonold and Fromme, 2014*). We confirmed and extended that result by showing that Kex2-GFP both arrived and departed slightly before Sec7-mCherry, with the two kinetic traces typically offset by 5–20 s (*Figure 11A* and *Video 6*). Thus, a green spot that lacks red fluorescence could represent a Golgi cisterna that has acquired Kex2-GFP and will soon acquire Sec7-mCherry. Alternatively, a green spot that lacks red fluorescence could represent a non-Golgi compartment that contains Kex2-GFP and will never contain Sec7-mCherry. To determine the relative abundance of these two classes of Kex2-GFP-labeled structures, we analyzed *Video 6* and identified the green spots that lacked red fluorescence and that could be reliably tracked for at least 30 s after their initial appearance. A total of 26 such structures were detected. As shown in *Video 7*, all 26 structures subsequently acquired Sec7-mCherry. Additional Kex2-GFP-labeled structures that could not be

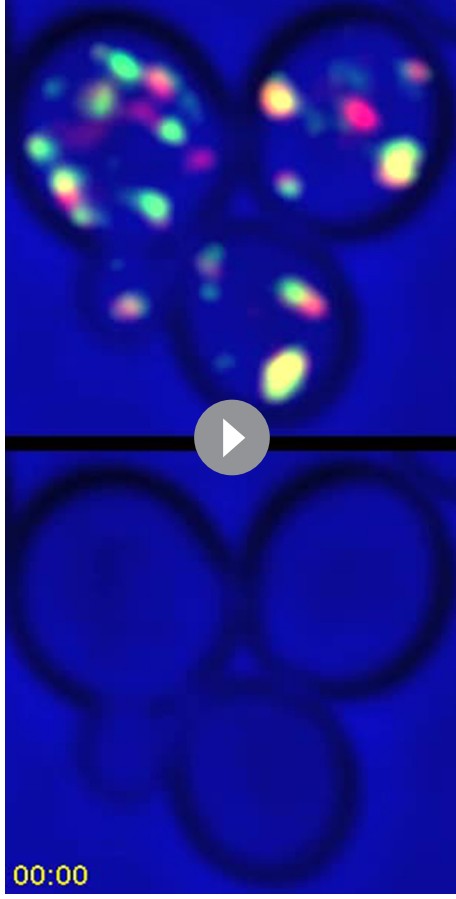

**Video 6.** Combined original and edited movie comparing the dynamics of Kex2 and Sec7. Cells expressing Kex2-GFP and Sec7-mCherry were imaged by dual-color 4D confocal microscopy. Scattered light images were recorded in the blue channel. Complete Z-stacks were collected every 2 s for 5 min, and the data were deconvolved, bleach corrected, and average projected. Edited movies tracking two representative cisternae were generated, merged, and appended below the original movie. See also *Figure 11A*.

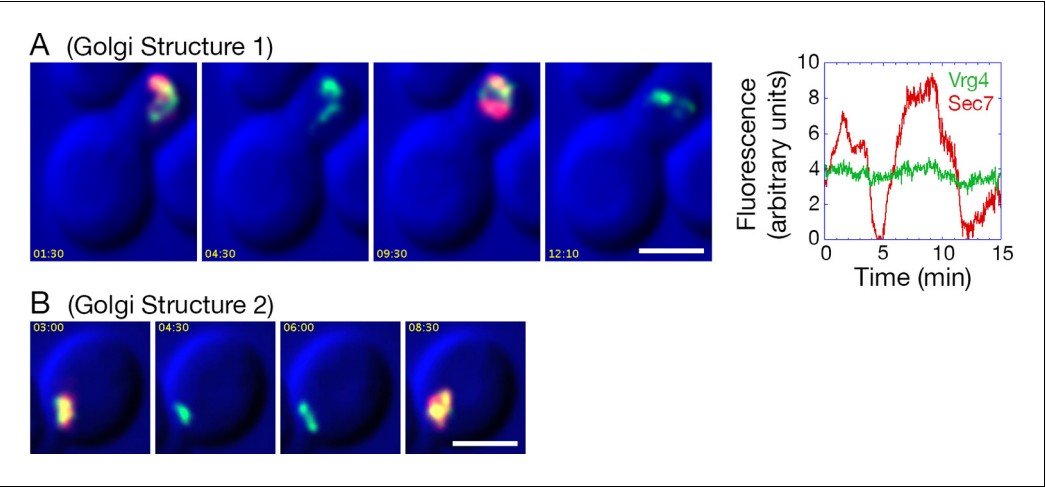

**Figure 9.** Effects of anchoring COPI on Golgi maturation dynamics. A strain expressing the OM45-FKBPx4 anchor as well as Sec21-FRBx2 was transformed to express the early Golgi marker GFP-Vrg4 and the late Golgi marker Sec7-DsRed. Logarithmically growing cells were attached to a coverglass-bottom dish, and were then treated with 1 µg/mL rapamycin for 10 min, followed by dual-color 4D confocal imaging to generate *Video 3*. Edited data sets were generated to analyze the two hybrid Golgi structures marked in *Figure 9—figure supplement 1*, yielding *Video 4* for Golgi Structure 1 and *Video 5* for Golgi Structure 2. Times are indicated in min:sec format. Scale bars, 2 µm. (**A**) Representative frames are shown from *Video 4*, and the green and red fluorescence intensities from Golgi Structure 1 are plotted versus time. (**B**) Representative frames are shown from *Video 5*.

The following figure supplement is available for figure 9:

**Figure supplement 1.** Sample frame from *Video 3*.

tracked for the full 30 s also acquired Sec7-mCherry (*Figure 11—figure supplement 1*). Therefore, most or all of the structures that label strongly with Kex2-GFP are late Golgi cisternae. The straightforward interpretation of these data is that Kex2 recycles within the Golgi, with kinetics slightly offset from those of Sec7.

Given that Sec7 arrives at a cisterna as Vrg4 is departing (*Losev et al., 2006*), we predicted that GFP-Vrg4 and Kex2-mCherry would show green-to-red maturation with a brief period of overlap. Although Kex2-mCherry gave a comparatively weak signal that made the analysis challenging, maturation from GFP-Vrg4 to Kex2-mCherry was indeed observed (*Figure 11B* and *Video 8*).

These control experiments set the stage for testing the effects of anchoring COPI. When COPI was inactivated, the dynamics of Kex2-mCherry were similar to those of Sec7-DsRed, with the levels of Kex2-mCherry alternately rising and falling in Golgi structures that were marked continuously by GFP-Vrg4 (*Figure 11C* and *Video 9*). We conclude that not only peripheral membrane proteins, but also transmembrane proteins of the late Golgi can recycle independently of COPI.

The combined data suggest that COPI inactivation selectively blocks recycling of early Golgi proteins. Thus, instead of an early Golgi cisterna maturing into a late cisterna, an early cisterna matures into a hybrid structure, which eventually loses its late Golgi proteins and then begins the process anew.

## Discussion

Thermosensitive yeast mutants are versatile tools for studying the secretory pathway (*Duden and Schekman, 1997*), but such a mutant has disadvantages. The molecular basis of the thermosensitivity is typically unknown, creating uncertainty about whether the mutant protein has been completely inactivated by the temperature shift. Moreover, a thermosensitive mutation may have detrimental effects even at the permissive temperature, in which case cellular functions will be compromised before the experiment begins. Both of these issues have been encountered with thermosensitive COPI mutants. For example, some of the commonly used COPI mutants show variable and relatively

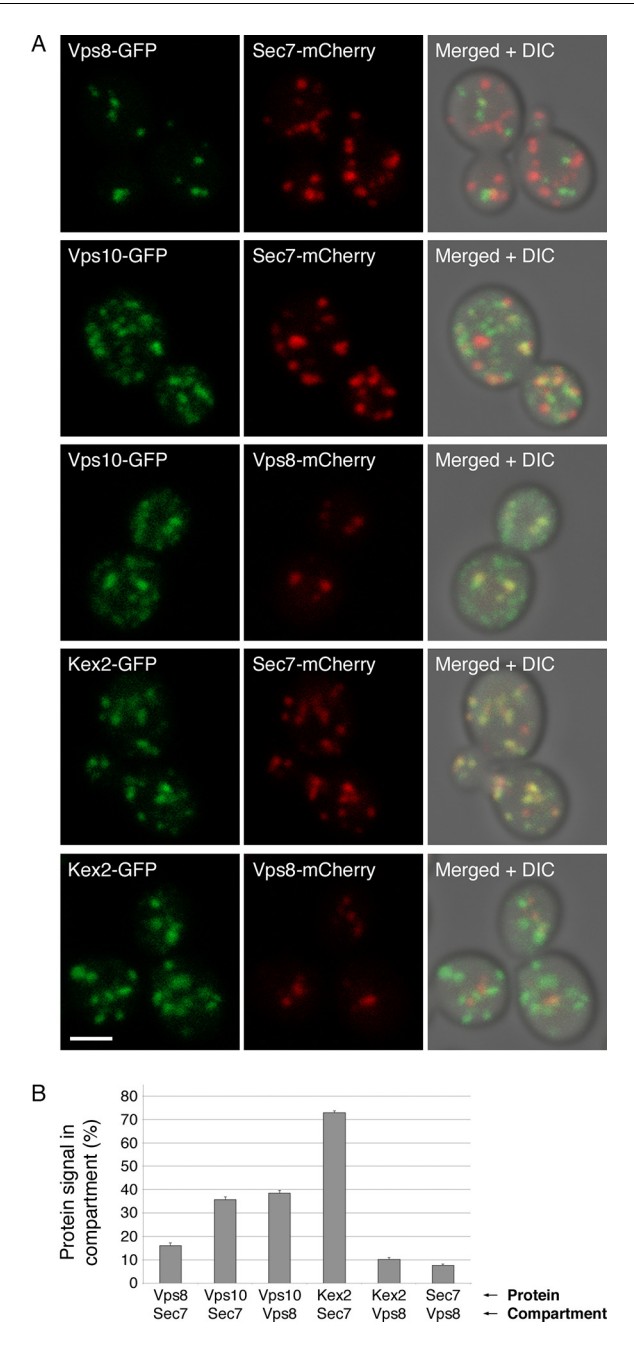

**Figure 10.** Distinct localization patterns of tagged Vps10 and Kex2. (**A**) Vps10-GFP localizes to both the Golgi and prevacuolar endosomes whereas Kex2-GFP is largely restricted to the Golgi. Late Golgi compartments were tagged with Sec7-mCherry, or else prevacuolar endosome compartments were tagged with Vps8-mCherry. The localizations of the Vps10-GFP and Kex2-GFP proteins were then examined with reference to these two compartments. As a control, Vps8-GFP was expressed together with Sec7-mCherry to confirm that the two compartments were separate. Representative projected confocal images are shown. Scale bar, 2 µm. (**B**) Quantitation of the data from (**A**). To analyze an image, a mask was created from the punctate compartment signal, and the percentage of the protein signal visible through the mask was then measured (*Levi et al., 2010*). For each strain, 40 images with ~2–4 cells per image were quantified. Bars represent mean percentage values with s.e.m. Based on the analysis of Vps8 and Sec, the background signal due to chance overlap in this assay was approximately 8–16%.

The following figure supplement is available for figure 10:

**Figure supplement 1.** Example of an unusual cell with some Kex2-GFP visible in prevacuolar endosomes.

weak phenotypes (*Gaynor and Emr, 1997*). That issue was addressed by isolating the strong *sec21-3* mutation (*Gaynor and Emr, 1997*), but in our hands, Golgi morphology was perturbed in *sec21-3* cells even at the permissive temperature. Thus, thermosensitive mutants have not been ideal for examining the role of COPI in yeast.

We addressed this problem by using the anchor-away method. Growth tests indicated that wild-type FRB is suitable as a tag for gene replacement while the destabilized FRB(T2098L) mutant is not. With regard to anchors, our initial trials employed an FKBPx2-tagged version of the plasma membrane protein Pma1 (*Haruki et al., 2008*), but strains expressing Pma1-FKBPx2 had growth defects and were genetically unstable. Better results were obtained with the ribosomal Rpl13A-FKBPx2 anchor (*Haruki et al., 2008*). We also generated a mitochondrial OM45-FKBPx4 anchor, which is effective in combination with an FRBx2 tag. Control experiments indicated that these versions of the anchor-away system allow COPI to be inactivated quickly and reliably, and can therefore serve to complement thermosensitive mutants for studying COPI function. COPII can also be inactivated with the anchor-away system, although the response is slightly different than for COPI, indicating that this system needs to be tested for each component that is being inactivated.

It should be noted that "anchor-away" is something of a misnomer here because the anchored COPI remained associated with Golgi membranes. Thus, when OM45-FKBPx4 was used as the anchor, entire Golgi compartments apparently became tethered to mitochondria. Such an effect is not unexpected because Golgi cisternae are mobile in the yeast cytoplasm (*Wooding and Pelham, 1998*; *Losev et al., 2006*). Despite retaining its association with Golgi membranes, the anchored COPI was no longer functional as judged by inhibition of cell growth and of Golgi-to-ER recycling, implying that this method is suitable for studying the roles of COPI.

Our data confirm the earlier conclusion that general secretion is arrested by inactivating COPII but is only partially inhibited by inactivating COPI (*Gaynor and Emr, 1997*). However, instead of classifying secretory proteins in a binary fashion as being either COPI-dependent or -independent, we propose that COPI

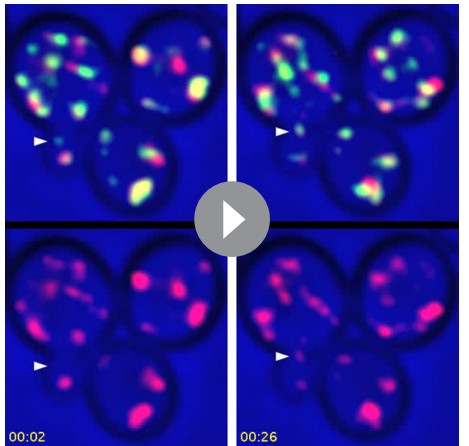

**Video 7.** Frame pairs from *Video 6* showing the consistent appearance of Sec7 in Kex2-containing structures. *Video 6* was analyzed to identify structures that labeled for Kex2-GFP but not Sec7-mCherry, and that could be followed for at least 30 s after becoming visible. Each frame pair highlights a single Kex2-GFP-containing structure at two closely spaced time points. In the left or right half of the frame pair, the merged green and red fluorescence signals are at the top and the red fluorescence signal is at the bottom. The arrowheads in the left half of the frame pair indicate the structure prior to the appearance of Sec7-mCherry, and the arrowheads in the right half of the frame pair indicate the same structure after the appearance of Sec7-mCherry. For convenience, the 26 frame pairs corresponding to distinct Kex2-GFP-containing structures are displayed as frames in a single movie. See also *Figure 11A* and *Figure 11—figure supplement 1*.

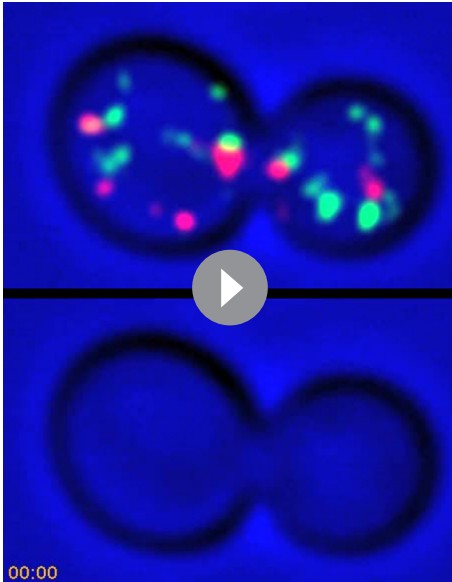

**Video 8.** Combined original and edited movie showing the dynamics of GFP-Vrg4 and Kex2-mCherry. Cells expressing GFP-Vrg4 and Kex2-mCherry were analyzed as in *Video 6*, except that the duration of the movie was 7 min. See also *Figure 11B*.

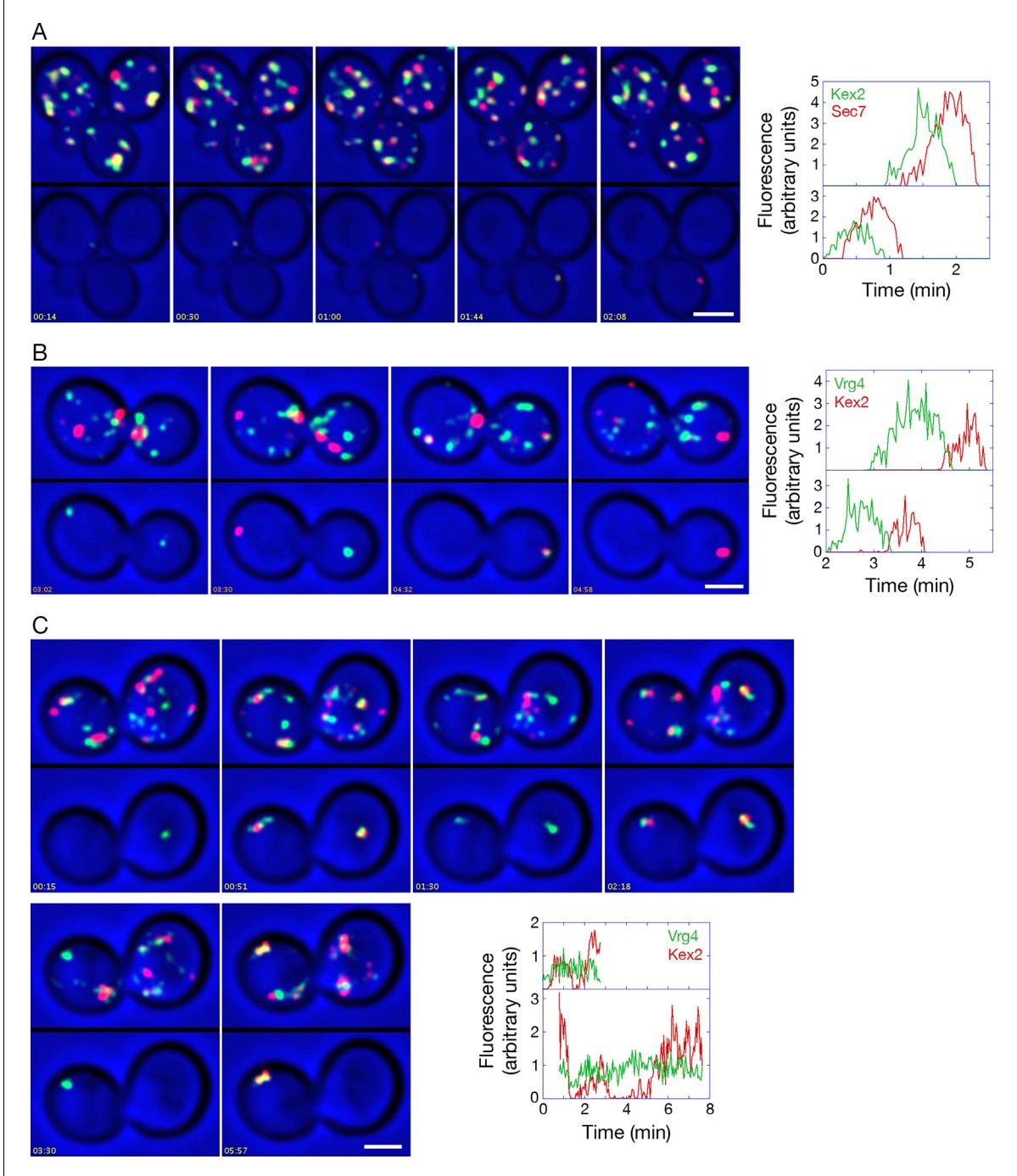

**Figure 11.** Analysis of Kex2 maturation dynamics with functional or inactivated COPI. (**A**) Kex2 maturation dynamics are slightly offset from those of Sec7. Cells expressing Kex2-GFP and Sec7-mCherry were attached to a coverglass-bottom dish and imaged by dual-color 4D confocal microscopy to generate *Video 6*, in which the top panel is the unedited movie and the bottom panel was generated from edited data sets used to quantify the fluorescence intensities from two cisternae. Representative frames from *Video 6* are shown together with the quantitation. Plotted in the bottom panel is fluorescence from the cisterna in the cell at the upper left, and plotted in the top panel is fluorescence from the cisterna in the cell at the lower right. Scale bar, 2 μm. (**B**) GFP-Vrg4 departs as Kex2-mCherry arrives during cisternal maturation. Imaging was performed as in (**A**), and representative frames from *Video 8* are shown together with the quantitation. Plotted in the bottom panel is fluorescence from the cisterna in the cell at the left, and plotted in the top panel is fluorescence from the cisterna in the cell at the right. Scale bar, 2 μm. (**C**) After COPI inactivation, GFP-Vrg4 persists in Golgi structures while Kex2-mCherry cycles. Imaging was performed as in (**A**), and representative frames from *Video 9* are shown together with the quantitation. Plotted in the bottom panel is fluorescence from the Golgi structure in the cell at the left, and plotted in the top panel is fluorescence

*Figure 11 continued on next page*

*Figure 11 continued*

from the Golgi structure in the cell at the right. These two Golgi structures were tracked for as long as they could be resolved from other fluorescent structures. Scale bar, 2 μm.

The following figure supplement is available for figure 11:

**Figure supplement 1.** Additional examples showing the appearance of Sec7 in Kex2-containing structures.

inactivation has a spectrum of effects that range from mild to severe depending on the protein. Among the proteins whose traffic was reported to be severely reduced in *sec21-3* cells at 37°C were the α-factor and carboxypeptidase Y precursors (*Gaynor and Emr, 1997*), both of which rely on the ER export receptor Erv29, which recycles from the Golgi to the ER (*Dancourt and Barlowe, 2010*). A plausible interpretation is that for certain secretory proteins, preventing COPI-dependent recycling of the cognate ER export receptors strongly inhibits traffic (*Gaynor and Emr, 1997*).

This line of reasoning raises a question: how can the secretory pathway operate at all after a block in COPI-dependent recycling, given that ER-to-Golgi traffic relies on SNARE proteins that are retrieved to the ER by COPI (*Barlowe and Miller, 2013*)? We propose that when COPI is inactivated, the cell replenishes ER-to-Golgi SNAREs through new protein synthesis. Other components such as ER export receptors will also be replenished, but at varying rates depending on their synthesis kinetics. According to this model, COPI inactivation will have the following effects on ER export. (a) COPII vesicle production will be slowed but not halted. (b) For a given secretory protein, the rate of ER export will depend on how rapidly the cognate ER export receptor is replenished and/or how efficiently the protein is packaged into COPII vesicles in the absence of an ER export receptor.

After a secretory protein leaves the ER, COPI is thought to help drive traffic through the Golgi, yet when yeast COPI is inactivated, proteins can still be secreted. Insight into this puzzle came from fluorescence microscopy. Soon after COPI is inactivated, early and late Golgi proteins change from marking separate compartments to associating with one another in hybrid Golgi structures. These hybrid structures are very dynamic, and we suspect that the compartments containing early and late Golgi markers exchange material, although our analysis cannot determine whether this exchange involves transient fusion events or other types of transport intermediates. In any case, the hybrid Golgi structures have early Golgi character, so secretory proteins can presumably reach these structures in COPII vesicles, and the hybrid Golgi structures show relatively normal cycling of late Golgi components, so secretory proteins can presumably depart to the cell surface in transport carriers. Compared to the unperturbed Golgi, the hybrid Golgi structures may yield altered glycosylation (*Gaynor and Emr, 1997*) but they are functional for membrane traffic. This analysis plausibly explains how yeast cells can continue to secrete after COPI inactivation.

While it is interesting to explore the effects of COPI inactivation, the larger goal is to understand the normal role of COPI in Golgi traffic. We have proposed that Golgi maturation occurs in discrete stages, and that the transition from the "carbohydrate synthesis" stage to the "carrier formation" stage involves the COPI-dependent

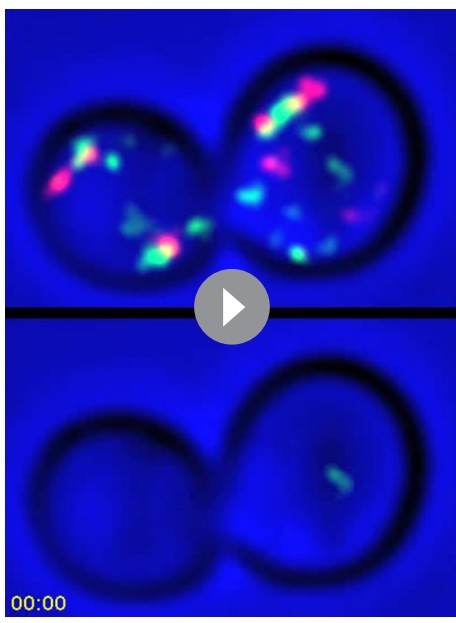

**Video 9.** Combined original and edited movie showing the dynamics of GFP-Vrg4 and Kex2-mCherry after anchoring COPI. Cells in which Sec21 had been anchored to mitochondria were imaged as in *Video 7* to visualize the dynamics of GFP-Vrg4 and Kex2-mCherry, except that the duration of the movie was 10.5 min and the interval between Z-stacks was 3 s. See also *Figure 11C*.

recycling of resident Golgi proteins to younger cisternae (*Day et al., 2013*; *Papanikou and Glick, 2014*). In yeast, Vrg4 is a marker of the carbohydrate synthesis stage while Sec7 is a marker of the carrier formation stage. As a cisterna acquires Sec7, it loses Vrg4. Loss of Vrg4 probably occurs by COPI-mediated transport, based on genetic, biochemical, and electron microscopic evidence that Vrg4 is recycled in COPI vesicles (*Abe et al., 2004*; *Mari et al., 2014*). The implication is that if COPI is inactivated, a cisterna could acquire Sec7 while failing to lose Vrg4, thereby generating a hybrid compartment (*Figure 12A*).

We tested this idea by capturing 4D movies after COPI had been inactivated. As predicted, after COPI inactivation, Golgi structures acquired Sec7-DsRed while failing to lose GFP-Vrg4. This result provides the first direct evidence that COPI plays a role in cisternal maturation. Specifically, we conclude that COPI helps to drive cisternal maturation by recycling early Golgi proteins such as Vrg4 from maturing cisternae. Vrg4 could recycle to younger cisternae either by intra-Golgi traffic, or by Golgi-to-ER traffic followed by delivery to newly forming cisternae. To evaluate these possibilities, we note that when COPII was inactivated by anchoring to mitochondria, GFP-Vrg4 showed no ER accumulation of the type that would be expected if this protein frequently returned to the ER. The data therefore suggest that COPI recycles Vrg4 within the Golgi.

After a hybrid Golgi structure was generated by COPI inactivation, Sec7-DsRed frequently disappeared from the structure and subsequently reappeared. Thus, the recycling pathway for Sec7 remained functional after COPI inactivation. A likely mechanism for COPI-independent recycling of Sec7 is dissociation from the membrane of one cisterna followed by reassociation with the membrane of a younger cisterna (*Figure 12B*). Consistent with this model, Sec7 recruitment to the membrane is known to require activated GTPases, implying that GTP hydrolysis releases Sec7 into the cytosol for another round of recruitment (*McDonald and Fromme, 2014*).

Can late Golgi transmembrane proteins also recycle independently of COPI? Some late Golgi transmembrane proteins traffic to prevacuolar endosomes and back. An example is Vps10, which is delivered to prevacuolar endosomes in clathrin-coated vesicles with the aid of Gga adaptors (*Costaguta et al., 2001*; *Deloche et al., 2001*; *Abazeed and Fuller, 2008*). Recycling of Vps10 from prevacuolar endosomes to the Golgi is mediated by the retromer complex (*Seaman et al., 1997*; *Seaman, 2005*). This loop through the prevacuolar endosome is presumably independent of COPI (*Figure 12B*). Other late Golgi transmembrane proteins may recycle within the Golgi itself (*Wong and Munro, 2014*). Candidates for such an intra-Golgi recycling pathway include trafficking machinery proteins such as Tlg1 (*Valdivia et al., 2002*), phospholipid translocases such as Drs2 (*Liu et al., 2008*), proteins such as Chs3 and Pin2 that reside in the late Golgi before undergoing regulated export to the plasma membrane (*Valdivia et al., 2002*; *Ritz et al., 2014*; *Spang, 2015*), and processing proteases such as Kex2 that act on secretory cargoes (*Fuller et al., 1988*). These proteins are often assumed to recycle from early endosomes to the late Golgi, but we found that at least some of the yeast compartments described as early endosomes are identical to the late Golgi (*Bhave et al., 2014*), suggesting that the proposed early endosome-to-Golgi recycling pathway could actually be an intra-Golgi recycling pathway. This intra-Golgi recycling pathway might or might not involve COPI.

To explore this issue, we focused on Kex2, which was described together with Sec7 as one of the first known markers of the yeast Golgi (*Franzusoff et al., 1991*; *Redding et al., 1991*). By fluorescence microscopy, Kex2 and Sec7 show substantial but incomplete overlap. The existence of structures that label with either Kex2 alone or Sec7 alone can now be explained, by our work plus a recent study (*McDonald and Fromme, 2014*), as being due to an offset in the kinetic behaviors of these two proteins during cisternal maturation. Kex2 arrives at a Golgi cisterna ~5–20 s before Sec7 and then departs ~5–20 s before Sec7. Thus, Kex2 apparently recycles within the Golgi somewhat ahead of Sec7. Our interpretation argues against the view that Kex2 resembles Vps10 in cycling between the Golgi and prevacuolar endosomes (*Abazeed et al., 2005*; *De et al., 2013*). Although we occasionally see Kex2-GFP in prevacuolar endosomes, this pool is very small, probably because Kex2 trafficking to prevacuolar endosomes reflects either a secondary recycling pathway or a degradation pathway. The proposed recycling of Kex2 within the Golgi merits further exploration. Meanwhile, we found that COPI inactivation did not prevent recycling of Kex2. This result, together with the finding that COPI is concentrated at the early Golgi, suggests that intra-Golgi recycling of late Golgi transmembrane proteins is independent of COPI.

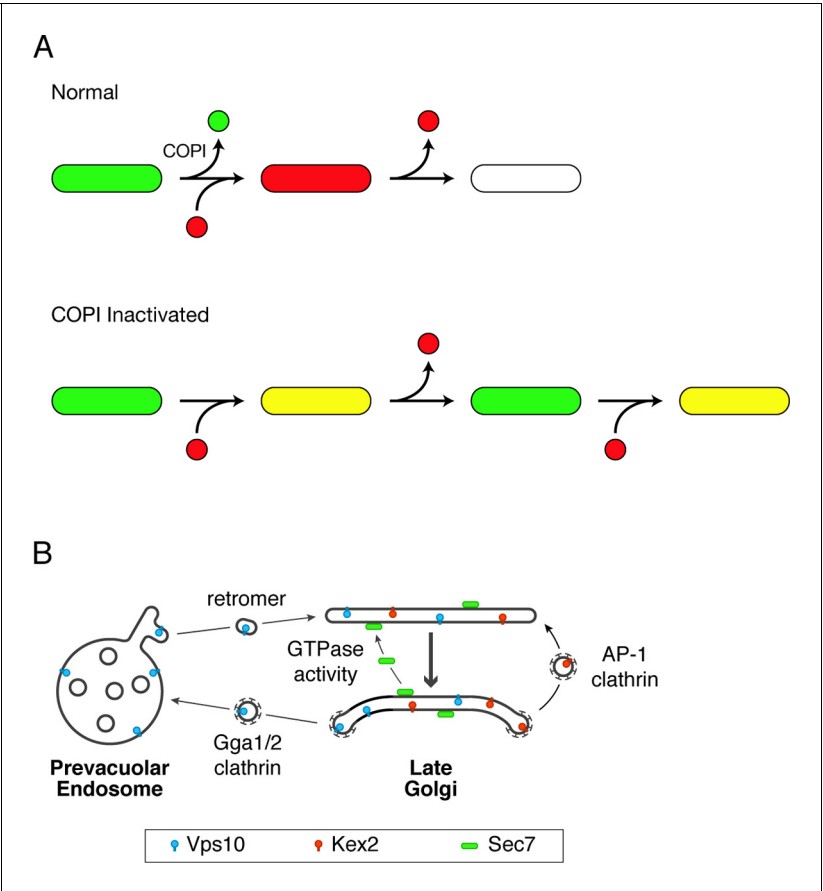

**Figure 12.** Working hypotheses for Golgi protein recycling by COPI-dependent and COPI-independent pathways. (**A**) The existence of multiple intra-Golgi recycling pathways can explain why inactivating COPI generates hybrid structures that repeatedly gain and lose late Golgi proteins. Green represents early Golgi proteins that recycle in COPI vesicles, and red represents late Golgi proteins that recycle by COPI-independent pathways. Under normal conditions, early Golgi proteins depart while late Golgi proteins arrive, and then late Golgi proteins depart in turn. When COPI is inactivated, the recycling of early Golgi proteins is inhibited, so when late Golgi proteins arrive, a hybrid structure is generated. This hybrid structure can subsequently lose late Golgi proteins by COPI-independent pathways, and then the process begins again. (**B**) During maturation of the late Golgi, proteins are likely to recycle by several pathways. The thick arrow represents cisternal maturation that occurs during and after conversion to a late Golgi compartment (*Daboussi et al., 2012*; *Day et al., 2013*). Peripheral membrane proteins such as Sec7 are recruited to late Golgi cisternae by activated GTPases, and are subsequently released from the membrane by GTP hydrolysis. Some transmembrane proteins such as Vps10 travel to the prevacuolar endosome in clathrin-coated vesicles with the aid of the Gga1 and Gga2 adaptors, and then recycle to newly formed late Golgi cisternae with the aid of cargo scaffolds such as retromer. Other transmembrane proteins such as Kex2 are postulated to recycle from older to younger late Golgi cisternae in clathrin-coated vesicles with the aid of the AP-1 adaptor.

A candidate for the carriers that recycle late Golgi transmembrane proteins is clathrin-coated vesicles containing the AP-1 adaptor (*Figure 12B*). Yeast AP-1 has been implicated in the late Golgi localization of multiple proteins including Tlg1, Drs2, Chs3, and Pin2 (*Valdivia et al., 2002*; *Foote and Nothwehr, 2006*; *Liu et al., 2008*; *Barfield et al., 2009*; *Myers and Payne, 2013*; *Ritz et al., 2014*). One study provided evidence that AP-1 also plays a role in Kex2 localization (*Abazeed and Fuller, 2008*). Those data have been thought to reflect a recycling pathway from early endosomes, but yeast AP-1 localizes to the late Golgi (*Daboussi et al., 2012*), supporting the alternative view that AP-1 mediates recycling from older to younger cisternae at a late stage in Golgi maturation (*Valdivia et al., 2002*; *Liu et al., 2008*). We speculate that Kex2 recycling involves AP-1, which might act in conjunction with the co-adaptor Ent5 (*Costaguta et al., 2006*; *Copic et al., 2007*; *Daboussi et al., 2012*). In mammalian cells, the functions attributed to AP-1 include retrograde transport from recycling endosomes and TGN-derived transport carriers (*Hirst et al., 2012*;

*Bonifacino, 2014*; *Matsudaira et al., 2015*), and those pathways are potentially analogous to recycling within the late Golgi of yeast.

AP-1-containing retrograde vesicles may be captured at the yeast Golgi by effectors of Ypt6, a Rab GTPase that is recruited prior to Sec7 (*Suda et al., 2013*). One Ypt6 effector is the GARP complex, which has been implicated in endosome-to-Golgi recycling (*Bonifacino and Hierro, 2011*) but could also play a similar role in intra-Golgi recycling. Another Ypt6 effector is the Sgm1 tether (*Siniossoglou and Pelham, 2001*), and intriguingly, the homologous TMF tether in mammalian cells captures intra-Golgi transport carriers carrying late Golgi proteins (*Wong and Munro, 2014*). The roles of Ypt6 effectors and AP-1 in late Golgi recycling can be tested in yeast using approaches like the ones described here.

Our working hypothesis is that yeast Golgi maturation involves distinct pathways that act in sequence. At the early Golgi, transmembrane proteins are recycled by intra-Golgi COPI vesicles. At the late Golgi, proteins are recycled by multiple mechanisms that involve either transit through the cytosol, or traffic to prevacuolar endosomes and back, or retrograde transport from older to younger cisternae (*Figure 12B*). An open question is how these various pathways are coordinated to maintain organellar homeostasis.

When COPI is inactivated, Golgi compartmentation is lost, and when Golgi compartmentation is lost, COPI is no longer needed for secretion. We interpret these findings to mean that Golgi compartmentation and COPI-driven cisternal maturation are aspects of the same phenomenon. Although Golgi compartmentation is broadly conserved, and is thought to provide fine control of glycan assembly while keeping secretory cargoes in the organelle long enough for complete processing (*Stanley, 2011*; *Ruiz-May et al., 2012*; *Day et al., 2013*), some organisms such as microsporidia forgo Golgi compartmentation and rely instead on a fused Golgi network (*Beznoussenko et al., 2007*; *Takvorian et al., 2013*). Yet COPI components are present in microsporidia (*Beznoussenko et al., 2007*; *Mowbrey and Dacks, 2009*), perhaps because an efficient secretory pathway always requires Golgi-to-ER recycling, even with a non-compartmentalized Golgi. In this view, Golgi-to-ER recycling is the core function of COPI, and many eukaryotes have adapted COPI for the additional purpose of maintaining separate Golgi compartments through cisternal maturation.

## Materials and methods

### Strains and plasmids

Experiments were done with derivatives of the haploid *S. cerevisiae* strain JK9-3da, which carries the mutations *leu2-3,112 ura3-52 rme1 trp1 his4* (*Kunz et al., 1993*). Yeast cells were grown in rich glucose medium (YPD), or in minimal glucose dropout medium (SD) (*Sherman, 1991*) or nonfluorescent minimal glucose dropout medium (NSD) (*Bevis et al., 2002*), with shaking at 200 rpm in baffled flasks. Growth media were obtained from Difco Laboratories (Detroit, MI, USA).

The *TOR1-1* mutation was introduced using the pop-in/pop-out method for gene replacement (*Rothstein, 1991*; *Rossanese et al., 1999*). To delete the *FPR1* gene, the *kanMX* cassette was amplified with the following primers to append sequences flanking the *FPR1* open reading frame: AACTCGAGTATAAGCAAAAATCAATCAAAACAAGTAATACGTACGCTGCAGGTCGAC and TAAAAAGCAGAAAGGCGGCTCAATTGATAGTACTTTGCTTATCGATGAATTCGAGCTCG. The resulting fragment was transformed into cells, which were plated on YPD containing 250 µg/mL G418 (Sigma-Aldrich, St. Louis, MO, USA) to select for double-crossover replacement of *FPR1*. A similar method was used to delete the *RER1* gene by replacing it with a hygromycin resistance cassette from pAG32 (*Goldstein and McCusker, 1999*).

The mitochondrial matrix was labeled by transforming cells with a centromeric plasmid that drove expression from the constitutive *ADH1* promoter of a mitochondrially targeted fluorescent protein, either mCherry in the case of pHS12-mCherry (*Sesaki and Jensen, 1999*; *Bevis and Glick, 2002*), or TagBFP in the case of p416-ADH::mito-TagBFP (*Murley et al., 2013*), which was obtained from Laura Lackner. To create a mitochondrial anchor, the gene encoding OM45 (*Yaffe et al., 1989*) was inserted between the strong constitutive *TPI1* promoter and the *CYC1* terminator in a vector that was integrated at the *TRP1* locus. The Sec71TMD-EGFP construct was subcloned from a plasmid provided by Ken Sato into the integrating vector YIplac128 (*Gietz and Sugino, 1988*). For video

microscopy, an integrating vector was used to overexpress Sec7-DsRed.M1x6 (*Losev et al., 2006*). All other tags were introduced by using pop-in/pop-out gene replacement to express proteins at endogenous levels, except that to boost the signal for Kex2-mCherry, a second copy of this construct was expressed from a centromeric plasmid. For GFP tagging, the variants used for gene replacement were either the monomeric mEGFP (*Zacharias et al., 2002*) or the monomeric superfolder msGFP (*Fitzgerald and Glick, 2014*). The FKBP and FRB genes were obtained from Ariad Pharmaceuticals (Cambridge, MA), and the MBP gene was obtained from New England Biolabs (Ipswich, MA). The mCherry gene (*Shaner et al., 2004*) was obtained from Roger Tsien (University of California at San Diego), and was modified at the N- and C-termini to create the mCherry2B variant used here.

DNA manipulations were simulated and recorded using SnapGene software (GSL Biotech, Chicago, IL). Annotated sequence files for 34 of the plasmids used in this study are included as a zip archive (*Supplementary file 1*), and can be opened with the free SnapGene Viewer (http://www.snapgene.com/products/snapgene_viewer/).

## Fluorescence microscopy

To minimize the background signal for fluorescence microscopy, yeast cultures were grown in SD or NSD medium. Static images were captured with living cells that were compressed beneath a coverslip without fixation and then immediately viewed, and 4D data sets were acquired with cells attached to a concanavalin A-coated coverglass-bottom dish containing NSD medium (*Losev et al., 2006*). To capture static images by widefield microscopy, we used an Axioplan2 epifluorescence microscope (Zeiss, Thornwood, NY) equipped with a 1.4-NA 100x Plan Apo objective and a digital camera (Hamamatsu, Skokie, IL). To capture static images by confocal microscopy, we used either an SP5 (Leica, Buffalo Grove, IL) or an LSM 710 (Zeiss) scanning confocal microscope to collect Z-stacks, with pixel sizes of 50-60 nm and Z-step intervals of ~0.3 μm. To capture 4D confocal movies with two fluorescence channels (red and green) and a scattered light channel (blue), cells were imaged at room temperature essentially as previously described (*Losev et al., 2006*), except that we used an SP5 microscope with pixel sizes of 50–60 nm and Z-step intervals of 0.29 μm to collect ~20–24 optical sections every 2 s. For the strain expressing GFP-Vrg4 and Kex2-mCherry, the pixel size was increased to 80 nm, the Z-step interval was increased to 0.34 μm, and the interval between Z-stacks for rapamycin-treated cells was increased to 3 s.

Some of the static confocal images and all of the 4D confocal data were deconvolved using Huygens software (Scientific Volume Imaging, Hilversum, The Netherlands). For the strain expressing GFP-Vrg4 and Kex2-mCherry, a single pass with a 2D hybrid median filter (*Hammond and Glick, 2000*) was performed to smooth the data before deconvolution. Adobe Photoshop and ImageJ (http://rsbweb.nih.gov/ij/) were used to colorize and merge the images, adjust brightness, and create average projections. Correction for exponential photobleaching was performed with an ImageJ plugin (http://cmci.embl.de/downloads/bleach_corrector). Editing and quantitation of 4D data sets was performed using custom plugins for ImageJ. These plugins allowed a hyperstack to be hybrid median filtered, converted to a montage time series, edited to remove extraneous fluorescence signals, converted back to a hyperstack, and quantified to measure red and green fluorescence intensities. A zip archive *Supplementary file 2* provides detailed instructions for capturing and processing 4D movies, together with our custom ImageJ plugins.

## Pulse-chase analysis

Cells were grown to log phase ($OD_{600}$ = 0.5–0.8) in SD medium. Prior to labeling, the cells were collected on a bottle-top filter, washed with SD lacking methionine (SD – Met), then resuspended in SD – Met at a concentration of 5 $OD_{600}$ units/mL. The concentrated cells were incubated for 30 min. For pulse labeling, 25 μCi of TRAN$^{35}$S-LABEL (MP Biomedicals, Santa Ana, CA) was added per $OD_{600}$ unit of cells, and the cells were incubated for 10 min. A 30-min chase was initiated by adding a 10x solution to give a final concentration of 5 mM unlabeled methionine plus 2 mM unlabeled cysteine. All of these manipulations were carried out with constant aeration at 30°C, except in the case of *Figure 6*, for which cells were grown at room temperature and then shifted to 37°C for the SD – Met preincubation, pulse, and chase. After the chase, cells were separated from medium by centrifugation at 5000 rpm (2300xg) in a microcentrifuge.

To analyze secreted proteins, the medium was adjusted to a final concentration of 10% trichloro-acetic acid (TCA). This mixture was incubated for 5 min at 60°C. TCA-precipitated material was collected by centrifugation for 5 min at full speed (16,000xg) in a microcentrifuge, then solubilized by vigorous vortexing in 50 µL of SDS-PAGE sample buffer supplemented with 0.1 M dithiothreitol and 50 mM $Na^+$ PIPES, pH 7.5. The sample was incubated for 30 min at 37°C, followed by a 3-min spin at full speed in a microcentrifuge to remove insoluble material.

To analyze cellular proteins, the cell pellet was resuspended in the original volume of medium, and then TCA was added to a final concentration of 10%. This mixture was incubated for 5 min at 50°C followed by 5 min on ice. TCA-precipated material was collected by centrifugation for 5 min at 3000 rpm (1000xg) in a microcentrifuge, then resuspended in SDS-PAGE sample buffer supplemented with 0.1 M dithiothreitol and 50 mM $Na^+$ PIPES, pH 7.5. The sample was incubated for 30 min at 60°C, followed by a 3-min spin at full speed in a microcentrifuge to remove insoluble material.

Each gel lane was loaded with a sample of secreted or cellular proteins corresponding to 0.25 $OD_{600}$ units of cells. SDS-PAGE was performed with Mini-PROTEAN TGX Tris/glycine 4–20% gradient gels using the Precision Plus Protein Dual Color Standards molecular weight markers (Bio-Rad, Hercules, CA). Gels were dried, and radioactive signals were detected using a Storm 860 molcular Imager (Molecular Dynamics, Sunnyvale, CA).

## Correlative fluorescence microscopy and electron tomography

A 100-mL culture of untreated or rapamycin-treated yeast cells was grown in SD to mid-log phase at 30°C with shaking. Cells were then concentrated by vacuum filtration using a 0.22-µm bottle-top filter (EMD Millipore, Billerica, MA). The cell paste was transferred to planchettes (Ted Pella, Redding, CA), cryo-fixed using a Bal-Tec HPM 010 high-pressure freezing machine (RMC, Tucson, AZ), and placed immediately into cryo-tubes containing a frozen cocktail of 0.1% uranyl acetate in anhydrous acetone. Samples were freeze substituted at -80°C for 48–60 hr in an EM AFS2 freeze substitution device (Leica). The temperature was then raised to -50°C and the samples were washed three times with acetone, followed by successive increasing overnight infiltrations with Lowicryl K4M resin (25, 50, 75, and 100%), followed by three incubations of 1 hr each with 100% resin. Infiltrated samples were placed in molds and polymerized with ultraviolet light at -50°C for 13 hr. To preserve fluorescence, the plastic blocks were stored at -20°C prior to sectioning. Sections of 300–1500 nm were cut with an EM UC6 ultramicrotome (Leica) and placed on 200 mesh carbon-formvar coated London-Finder copper grids (Electron Microscopy Sciences, Hatfield, PA).

For fluorescence microscopy, a grid was placed on a glass slide with the resin side up, and a 22x22 mm No. 1.5 glass coverslip with a 10-µl droplet of 500 mM $Na^+$-HEPES, pH 7.5 was inverted onto the grid. The coverslip was immediately sealed with wax. Imaging was performed with an LSM 710 confocal microscope. Dual-color Z-stacks were captured with a 1.4-NA 100x Plan-Apo oil objective using a pinhole of 1.2 Airy units and with voxels ranging in each dimension from 0.30 to 0.43 µm. The grid was then retrieved for analysis by electron microscopy.

For transmission electron microscopy (TEM) as well as TEM tomography and scanning transmission electron microscopy (STEM) tomography, images were collected on a Tecnai G2 F30 electron microscope (FEI, Hillsboro, OR) with a Schottky field-emission gun operating at 300 kV. Grids were prepared for either TEM or STEM tomography by floating each side of the grid for 10 min on a 10-µl drop of 15 nm colloidal gold bead solution (British BioCell International, obtained from Ted Pella). The samples were then stained for 8–15 min with 2% uranyl acetate, and placed into a Model 2040 dual-axis tomography holder (Fischione Instruments, Export, PA). For TEM tomography, 300–400 nm sections were analyzed using Serial EM (*Mastronarde, 2005*) to collect digital images at 15,000x magnification with a 4K UltraScan camera (Gatan, Pleasanton, CA) as a dual-axis tilt series over a range of -60° to +60° at tilt angle increments of 1°. For STEM tomography, the imaging conditions were as follows: extraction voltage = 4250 V, gun lens = 3, condenser aperture = 50 mm, and camera length range = 200–500 mm. Images were collected using a Model 3000 annular dark field detector (Fischione) placed above the viewing screen, and a Model 805 bright- and dark-field detector (Gatan) below the viewing screen. Images were collected as a dual-axis tilt series over a range of -60° to +60° at tilt angle increments of 1° using the FEI STEM tomography software. All tomograms

were reconstructed and analyzed using IMOD software (*Kremer et al., 1996*). The estimated resolution of the STEM tomograms is 8–12 nm (*Radermacher, 1992*).

## Acknowledgements

This work was supported by NIH grant R01 GM104010 and by funding through the Biological Systems Science Division, Office of Biological and Environmental Research, Office of Science, U.S. Dept. of Energy, under Contract DE-AC02-06CH11357. KJD was supported by NIH training grant T32 GM007183. Fluorescence microscopy was performed at the Integrated Microscopy Core Facility with the kind assistance of Vytas Bindokas, and electron microscopy was performed at the Electron Microscopy Core Facility, with support for both facilities coming from the NIH-funded Cancer Center Support Grant P30 CA014599.

## Additional information

### Funding

| Funder | Grant reference number | Author |
| --- | --- | --- |
| National Institutes of Health | R01 GM104010 | Benjamin S Glick |
| U.S. Department of Energy | DE-AC02-06CH11357 | Benjamin S Glick |
| National Institutes of Health | T32 GM007183 | Kasey J Day |

The funders had no role in study design, data collection and interpretation, or the decision to submit the work for publication.

### Author contributions

EP, KJD, Conception and design, Acquisition of data, Analysis and interpretation of data, Drafting or revising the article; JA, Conception and design, Acquisition of data, Analysis and interpretation of data; BSG, Conception and design, Analysis and interpretation of data, Drafting or revising the article, Conception and design, Acquisition of data

## Additional files

### Supplementary files

• Supplementary file 1. A zip archive of 34 annotated plasmid sequence files for constructs used in this study.

• Supplementary file 2. A zip archive of three Microsoft Word files describing how yeast 4D confocal data are collected, processed, and analyzed, together with six relevant ImageJ plugins.

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
