## [Decision Letter]

[Editors’ note: a previous version of this study was rejected after peer review, but the authors submitted for reconsideration. The previous decision letter after peer review is shown below.]

Thank you for choosing to send your work entitled "COPI selectively drives maturation of the early Golgi" for consideration at *eLife*. Your full submission has been evaluated by Vivek Malhotra (Senior editor) and three peer reviewers, one of whom is a member of our Board of Reviewing Editors, and the decision was reached after discussions between the reviewers. Based on our discussions and the individual reviews below, we regret to inform you that your work will not be considered further for publication in *eLife*. While the reviewers lauded the high quality of your study, the consensus view was that the advances that it presents do not carry the significance required to merit publication in *eLife*. In addition, as described in the comments from Reviewer 3 (point 2), your argument that Kex2 recycles within the Golgi, rather than from the endosome to the Golgi, is undermined by a published report (Suda et al, 2013). The reviews are included in their entirety below.

*Reviewer #1:*

This is an investigation of the role of COPI in the secretory pathway. The authors have optimized the anchor-away technique as a means to rapidly inactivate COPI function in yeast cells in order to assess the consequences to the maturation of Golgi cisternae and to secretion. They report that the transition from an early cisterna, marked by GFP-Vrg4, to a late cisterna, marked by Sec7-GFP, fails upon inactivation of COPI. In contrast, a maturation event monitored by Kex2-GFP dynamics is not affected, leading the authors to conclude that COPI is required solely for recycling of early Golgi proteins.

This is a carefully executed, nicely presented study and I don't have any serious technical concerns, as much of the paper is an extensive validation of the anchor away approach. The significance of the major points of the work, however, is lessened by several factors.

1) Prior reports (esp. Todorow et al, 2000; Tu et al. Science, 2008; Tu et al. 2012) implicated a role for COPI in Golgi localization of multiple early/medial Golgi residents in yeast cells via recycling through the ER. Although this paper more directly visualizes a role for COPI for recycling of an early Golgi resident (Vrg4), the advance presented here is modest and questions regarding the role of COPI recycling pathway remain to be addressed. Does COPI direct Vrg4 to the ER, or to an earlier Golgi compartment? Related, Rizzo et al. (2013) provided evidence that an artificial Golgi resident based on mannosidase II (MAN-FM) recycles via COPI vesicles/tubules from the trans Golgi cisterna of HeLa cells. Is Vrg4 recognized by COPI?

2) As the authors point out, it has been established that multiple late Golgi proteins (including Kex2) rely on AP-1/clathrin to maintain Golgi residence, and that COPI localizes predominantly to early Golgi compartments, so the finding that COPI inactivation does not prevent removal of Kex2 from the hybrid Golgi compartment is not unexpected. Further, I don't find the argument that Kex2 recycles within the Golgi (rather than from endosomes to the Golgi) to be sufficiently substantiated.

3) This work confirms, and modestly elaborates, that secretion of some proteins continues in COPI temperature sensitive mutants (Gaynor and Emr, 1998), which was shown many years ago (as the authors point out).

*Reviewer #2:*

In this study, the authors test the effects of functional removal of the COPI machinery on protein secretion and cisternal maturation in yeast. Previous studies with thermosensitive COPI mutants had indicated that COPI ablation inhibits the secretion of certain proteins, but not of others, and that it slows down, but does not block, cisternal maturation.

They now use a potentially 'cleaner' and more radical approach based on anchoring away specific COPI subunits to revisit the same issues. Using this technique, they find that secretion is strongly inhibited, although to different extents for different proteins, and that the maturation process and the organization of the Golgi are markedly altered. In control cells, early cisternae lose their residents to newly forming cis elements and simultaneously acquire trans Golgi markers, which they later loose, presumably during formation of secretory post Golgi carriers. In cells where COPI is inactivated by the anchoring away approach, early and late Golgi residents merge (partially) into a hybrid compartment, where they behave differently: the cis residents become unable to cycle and remain stably in this compartment (suggesting that they normally cycle through COPI) while trans residents can still cycle on and off the hybrid Golgi body though with delayed kinetics (suggesting that they cycle via a COPI- independent mechanism).

The authors interpret these data in the framework of the cisternal maturation scheme: because of the loss of COPI, the maturation process is inhibited at the cis side and cis-cisternae are 'stabilized'; they can thus serve as 'acceptors' of the trans residents, resulting in a hybrid Golgi. This hybrid compartment can still absorb input from the ER and release secretory material towards the surface, albeit at slow rates.

What are the new and important statements made by this study? One is that COPI is essential for recycling of cis residents. While the role of COPI in recycling from the Golgi to the ER was a relatively well established fact, this was not the case for the intra-Golgi stage. The other is that the cycling of trans Golgi residents is COPI-independent, and this represents a surprise. The mechanism of cisternal maturation is obviously an important issue in membrane biology and these findings represent a potentially significant advance.

The study is technically very thorough and the factual statements are convincing and well supported. We feel, however, that it has limitations, in that it leaves some essential question open and inconsistencies unresolved. Certain key issues should be addressed by suitable arguments and/or by additional experiments.

The main problem relates to the difference between the previous results based on thermosensitive mutants and the present ones based on the anchoring away approach. The anchoring method, as noted by the authors, results in the adhesion of the whole Golgi cisternae to the mitochondria, rather than in the 'removal' of the COPI machinery from Golgi cisternae. The results of this are hard to predict and may not always be congruent with the specific inactivation of the concerned proteins. It is conceivable that the strong block in recycling of the cis-Golgi residents might be due to extensive proximity/adhesion between Golgi and mitochondrial membrane, rather than to a specific block of the COPI function. This possibility undermines the author's central conclusions about the role of COPI in the recycling of cis residents and should be appropriately discussed or addressed experimentally. Some relevant questions and considerations and some possible approaches are suggested below:

1) A re-examination of the ultrastructure of the Golgi-mitochondria contact due to anchoring might help. How extensive are the contacting surfaces? Does the morphology of these contacts justify the above doubts about the effects of anchoring on recycling?

2) Could the anchoring of the Golgi membranes to mitochondria through a COP-unrelated cis-Golgi protein reproduce the recycling block?

3) The authors obtain similar results by anchoring COPI to ribosomes, which might be less likely than mitochondria to severely obstruct membrane exchanges at the Golgi surface. Exploiting and characterizing the ribosome data at the ultrastructural level might help resolve the issue.

4) Another approach might be to consider that while early Golgi markers are blocked, late markers keep cycling. How are they disposed with respect to the contacting Golgi and mitochondrial surfaces?

*Reviewer #3:*

This study from the Glick lab uses the anchor-away technique to inactivate the COPI complex in order to examine its role in Golgi maturation. The authors begin with a thorough analysis of the anchor-away method for inactivating COPI proteins. The analysis is aided by comparisons to anchor-away inactivation of the COPII complex.

The most surprising finding is that secretion is not completely blocked when COPI is inactivated. Anchoring-away different COPI subunits results in similar secretion phenotype. The secretion phenotype also appears identical when comparing COPI inactivation by anchor-away versus ts-mutant (sec21-3). The authors show that anchor-away is preferable to thermosensitivity because at permissive temperature the sec21-3 mutant appears compromised (Golgi morphology defects), meaning that indirect effects may be observed when using this mutant.

Anchor-away is used to show that Golgi dynamics are dramatically altered upon inactivation of COPI, although post-Golgi recycling appears to still occur. In wild-type cells, it is shown that the Kex2 transmembrane protein, a cargo that cycles between the Golgi and endosomes, normally appears at Golgi compartments after the peak of the Vrg4 transmembrane protein, a marker for early Golgi compartments. Kex2 remains present while Sec7 appears at that compartment, and Kex2 then leaves the compartment before the Sec7 signal completely disappears. However, in COPI-inactivated cells there are two major differences: First, the morphology of the compartments are quite different, with early and late markers showing partial colocalization to larger "hybrid" compartments. Second, in mutant cells maturation is altered so that compartments labeled with an early marker (Vrg4) persist for a longer time while later markers/cargos (Sec7 and Kex2) appear to cycle in and out of the "hyrbrid" compartment.

Given these results, the authors conclude that COPI is important for maturation and recycling of the early Golgi, but dispensable for late Golgi function. I agree that this overall conclusion is generally supported by the data, and I think this is a very important result that warrants publication in a journal like *eLife*. However, I have a couple of major concerns that need to be addressed before I would consider the manuscript to be suitable for publication:

1) One potential concern is that the anchored-away COPI proteins are still associated with the Golgi. Although cell growth is blocked by anchoring COPI, given the resolution of the growth curve data it is a formal possibility that partial COPI activity remains, and growth may not halt until after 1 or 2 more cell divisions. This might affect the interpretation of the data (especially the secretion phenotype), if partial COPI activity remains. The authors imply that because the sec21-3 mutant displays an identical secretion phenotype to the anchor-away COPI phenotype, it is likely that both methods result in complete inactivation of COPI. However, I think the authors need to demonstrate that the Golgi-ER COPI trafficking pathway is completely blocked by anchor-away inactivation of COPI. There are various "retrieval assays" in the literature. Another option would be to test by microscopy whether a protein that normally cycles between the ER and Golgi (i.e. a p24 protein or other cycling ER cargo receptor) is now exclusively found in the Golgi.

2) In the second half of the Discussion, the authors propose that AP-1 traffics Kex2 directly from late Golgi to early Golgi compartments. This would be a fairly radical idea, and would upend the current model that AP-1 traffics Kex2 from the Golgi to endosomes, and then Kex2 is retrieved from endosomes back to the Golgi by other machinery. Yet the only data provided specifically in support of this new model is the statement "our video microscopy revealed that most and perhaps all of the Kex2-labeled compartments acquire Sec7, suggesting that Kex2 recycles from older to younger cisternae within the Golgi". I disagree with both the data and interpretations contained in this statement. First, by watching their videos it appears that there are a few smaller Kex2 structures that may disappear without acquiring Sec7. Second, if we assume the first part of the statement is correct, this observation is consistent with Kex2 being retrieved from endosomes (where it may reside for only a few seconds), via vesicles, to Golgi compartments that have not yet acquired Sec7. To my knowledge, the precise Golgi compartment that fuses with endosomally-derived vesicles has not been firmly established in yeast, but the Nakano lab found that Ypt6 precedes Sec7 at the Golgi (PNAS 2013). As Ypt6 (together with the GARP complex) is primarily responsible for tethering/fusing vesicles from endosomes, this suggests Kex2 retrieval from endosomes to the Golgi occurs at a stage of Golgi maturation just prior to acquisition of Sec7. Therefore, it seems unfounded to propose a role for AP-1 in recycling cargos from late Golgi to early Golgi compartments.

[Editors’ note: what now follows is the decision letter after the authors submitted for further consideration.]

Thank you for resubmitting your work entitled "COPI selectively drives maturation of the early Golgi" for further consideration at *eLife*. Your revised article has been favorably evaluated by Randy Schekman (Senior editor) and three reviewers, one of whom is a Member of our Board of Reviewing Editors. The manuscript has been improved but there are some remaining minor issues that the reviewers feel should be addressed before acceptance, as outlined below:

The reviewers recommend that you address the following points with revisions to the text:

1) Reviewer 1 and Reviewer 2 raise the issue that the anchor away approach that you employed results in different outcomes for Golgi residents when compared to the sec21-3 mutant and a Vps74/Golph3 mutant. They feel that this should be directly addressed in the Discussion. The comments from Reviewer 1 and 2 (below) elaborate this point.

2) Since the arrested cisterna ('hybrid compartment') appears to remain competent to receive ER-derived material, and to export material, a brief discussion of the implications of this with regard to interpreting the results from both a cisternal maturation mechanism and a stable compartment mechanism of transport should be included in the Discussion. The comments from Reviewer 2 are most relevant for this point.

3) Can the authors clarify in the text which of the studies cited in the Discussion (tenth paragraph) demonstrated that AP-1 was involved in Kex2 trafficking and describe the Kex2-localization phenotype(s) seen previously by others in AP-1 mutant strains?

The entire reviews from each of the referees are included below to help you appreciate their suggestions. Addressing any other points raised in the reviews is left to your discretion.

*Reviewer #1:*

In light of the observation that the anchor away approach results in entire COPI-coated cisternae to be associated with mitochondria, a major point of discussion amongst the reviewers during review of the earlier version of the manuscript regarded the question, does the technical approach (anchor away) impinge on cisterna function independently of COPI? It is puzzling to me that in the sec21-3 mutant (permissive temperature) GFP-Vrg4 transits the Golgi and is sorted to the vacuole, but it remains in place (in the hybrid compartment) in anchor away cells. The reason may lie in incomplete inactivation of COPI in the sec21-3 mutant (the figure shows permissive temperature), but difference in outcome is striking and begs resolution. The only way that I can think of that might meaningfully address this would be to compare results with 'inert' anchor away targets. What, for example, happens when if FRB is appended to a Golgi resident (e.g., Vrg4), so that the cisterna is addressed to the mitochondrion, but COPI is not manipulated? Despite this shortcoming, I think that in its present form the manuscript makes an important contribution to the field, even though the message, as summarized in the title, may be open to continuing debate.

*Reviewer #2:*

In this study, the authors test the effects of inactivating the COPI complex on cisternal maturation and protein secretion in yeast. Previous studies with thermosensitive COPI mutants had indicated that COPI inactivation inhibits the secretion of many but not all proteins (Gaynor and Emr), and that it slows down, but does not block, cisternal maturation as visualized by live microscopy (Matsuura-Tokita and Nakano).

The authors attribute these inconclusive results to the fact that the available ts mutants can be either still partially active (Matsuura-Tokita and Nakano) or, when fully inactive, can markedly perturb the morphology of the Golgi (this manuscript), making it impossible to monitor directly the maturation process by live microscopy.

To resolve these uncertainties, they use the anchoring-away technique applied to specific COPI subunits. They find that COPI inactivation results in complete block of the recycling of the cis Golgi marker Vrg4, which now permanently labels the cis cisternae. This indicates that Vrg4 normally recycles through COPI. Moreover, they find that trans-Golgi residents (Sec7 and Kex2) partially merge with these stabilized cis cisternae, forming hybrid cis/trans Golgi bodies; and also that these trans markers cycle on and off the hybrid structures (though with somewhat delayed kinetics). This suggests that these markers cycle via a COPI-independent mechanism.

The study is technically very thorough and the factual statements are convincing and well supported. My comments will therefore concern only the novelty and the significance of the results.

As the authors emphasize, their data are the first direct demonstration that COPI is required for cisternal maturation. Evidence that COPI is required for maintaining the localization of Golgi residents (via Vps74/Golph3) has been available for some time (e.g., Tu and Banfield), but these previous observations are compatible both with the stable cisternae model and the cisternal maturation model. Thus, while not unexpected, these results are an important step towards establishing the molecular mechanism of cisternal maturation, and in principle deserve publication in a major journal such as *eLife*. They also raise the following questions and comments.

The stabilization of cis Golgi markers in cis cisternae probably results from the specific type of COPI inactivation that is induced by the anchoring-away approach, which causes COPI to remain bound to the cis Golgi and so possibly to retain cis markers in the cis cisterna. However, the stabilization of the cis Golgi it is not the only possible outcome of COPI inactivation, under the maturation model. One might imagine, for instance, that if COPI were inactivated in a different way, e.g., by rapid degradation, this might cause cis markers to move forward along with cargo proteins to reach the vacuole or the plasma membrane. This is actually what happens when the COPI-bound adaptor Vps74 is removed: the Golgi enzymes leave the Golgi and move to the vacuole (Tu-Banfield). This apparent discrepancy might disconcert the reader. I therefore feel that the above point should be discussed, even though it does not in my view undermine the conclusion that COPI is needed for cis Golgi maturation.

The authors appear to interpret the data in the framework of the cisternal maturation scheme. However, the observation that the hybrid compartment can still absorb input from the ER and release secretory material towards the surface, albeit at slow rates, is difficult to reconcile with the maturation model. Since the recycling of cis markers is inhibited, and the formation of a new cis cisterna (a necessary step in the cisternal maturation mechanism) requires such recycling, a logical suggestion deriving from these data would be that the hybrid Golgi incorporates cargo and membrane from the ER by a mechanism different from cisternal maturation. In my view, these results are consistent with the idea that the main intra-Golgi transport mechanism is cisternal maturation, but they also suggest the existence of secondary mechanisms that are used by some protein classes, and/or of backup mechanisms used by the system when maturation fails. Again, I feel that, in order to avoid confusing the reader in a field that is already rich in controversies, the authors should discuss these issues, which, incidentally, are referred to also in the Introduction.

As noted, we do not have comments worthy of mention regarding the (high) technical quality of the figures and in general of the data.

*Reviewer #3:*

This work convincingly clarifies the role of COPI, providing further evidence for a primary function in retrograde trafficking that is important for Golgi maturation. I think the authors now do a good job of explaining why some cargos are more sensitive to COPI inactivation than others, and how a basal level of secretion can be supported in the absence of COPI function. Furthermore, they have clarified the itinerary of Kex2. Therefore, I am in support of publication.

My only remaining concern is a relatively minor one, which I would expect could be addressed through small changes in the text: Can the authors clarify in the text which of the studies cited in the Discussion (tenth paragraph) demonstrated that AP-1 was involved in Kex2 trafficking and describe the Kex2-localization phenotype(s) seen previously by others in AP-1 mutant strains? In other words, what is the existing evidence in the literature that led to the model of AP-1 functioning in Kex2 traffic between the Golgi and endosomes? Furthermore, can the authors briefly discuss the established role of AP-1 in mammalian cells (i.e., papers from Bonifacino and other labs), and whether the authors think AP-1 has similar or different functions in yeast? I think explicitly addressing these points will help readers to better understand the insights from the current data in light of previous studies.

---

## [Author Response]

[Editors’ note: the author responses to the previous round of peer review follow.]

An issue with the initial submission was that we got ahead of ourselves by building the interpretations on additional data from ongoing work. The COPI results have inspired us to explore the relationship between endosomes and the Golgi, and we have obtained evidence that yeast cells lack an early endosome compartment that is distinct from the TGN. That story is still coming together, but last year we published a crucial result by showing that compartments previously termed early endosomes are actually identical to the Sec7-labeled TGN (M. Bhave et al., 2014, J. Cell Sci. 127:250-7). As described below, an implication is that many late Golgi proteins likely recycle within the Golgi. This unconventional idea is justified more thoroughly in the revised manuscript and is presented as a working hypothesis.

The revision also includes two new data figures. First, as suggested by a reviewer, we confirm that our COPI inactivation protocol inhibits Golgi-to-ER recycling. Second, because our inferences about Kex2 trafficking were a concern for the reviewers, we quantitatively characterize Kex2 localization and dynamics. The major contributions in the revised manuscript are as follows:

• We use 4D microscopy of GFP-Vrg4 to provide clear evidence that Golgi maturation requires COPI. Yeast 4D microscopy is the only method available for directly visualizing cisternal maturation. *This advance answers a central and long-standing question in the field.*

• We show that when COPI is functionally inactivated, the yeast Golgi reorganizes into hybrid structures that remain dynamic with respect to the cycling of late Golgi components. This result can explain the puzzling earlier observation, reproduced and extended here, that the yeast secretory pathway remains partly operational in the absence of COPI function.

• We provide a carefully controlled analysis of the anchor-away method, which has attracted considerable interest. Our data document the effects of various parameters as well as strengths and limitations of the method.

• We provide evidence that Kex2 cycles within the Golgi rather than between endosomes and the Golgi as is generally assumed. Although this conclusion will benefit from further validation, the present results establish Kex2 as a model protein for studying COPI- independent intra-Golgi recycling.

*Reviewer #1:*

*[…] This is a carefully executed, nicely presented study and I don't have any serious technical concerns, as much of the paper is an extensive validation of the anchor away approach. The significance of the major points of the work, however, is lessened by several factors.*

*1) Prior reports (esp. Todorow et al, 2000; Tu et al. Science, 2008; Tu et al. 2012) implicated a role for COPI in Golgi localization of multiple early/medial Golgi residents in yeast cells via recycling through the ER. Although this paper more directly visualizes a role for COPI for recycling of an early Golgi resident (Vrg4), the advance presented here is modest and questions regarding the role of COPI recycling pathway remain to be addressed. Does COPI direct Vrg4 to the ER, or to an earlier Golgi compartment? Related, Rizzo et al. (2013) provided evidence that an artificial Golgi resident based on mannosidase II (MAN-FM) recycles via COPI vesicles/tubules from the trans Golgi cisterna of HeLa cells. Is Vrg4 recognized by COPI?*

The papers mentioned by the reviewer are relevant and we now cite them in the Introduction. But while a role for COPI in the recycling of resident Golgi proteins has some experimental support, the COPI data from both yeast and mammalian studies have been confusing. For example:

• Yeast COPI subunits were originally implicated in anterograde ER-to-Golgi transport. That interpretation is no longer favoured, probably because the early papers reported indirect effects.

• Rizzo et al. did indeed provide evidence in 2013 that a resident Golgi protein could recycle from mammalian *trans*-Golgi cisternae as predicted by the maturation model, but a 2013 paper by Lavieu et al. (published in *eLife*) concluded that mammalian Golgi cisternae do not mature. The discrepancies between those parallel studies have yet to be resolved.

• Gaynor and Emr (1997) reported that COPI is dispensable for the secretion of some yeast proteins. To our minds, that paper has long been the most baffling in the field, because the experimental data were strong and yet the conclusion was seemingly at odds with all extant models for Golgi traffic.

Video microscopy of the yeast Golgi is currently the only direct assay for cisternal maturation, and our results with GFP-Vrg4 are the first direct evidence that maturation requires COPI. We believe that this advance is more than modest. Others in the field apparently agree that this sort of analysis is crucial because the Nakano lab is also trying to characterize how COPI inactivation affects Golgi dynamics.

Finally, an interaction of Vrg4 with COPI was nicely shown by Abe et al. (2004), and that evidence was one of our motivations for using Vrg4 as an early Golgi marker. The reviewer points out that we cannot say with certainty that Vrg4 recycles within the Golgi rather than returning to the ER. However, when we inactivated COPII by the anchor-away method, GFP- Vrg4 showed no ER accumulation (see Figure 7), implying that Vrg4 does not cycle rapidly back to the ER. We now discuss this point in the text.

*2) As the authors point out, it has been established that multiple late Golgi proteins (including Kex2) rely on AP-1/clathrin to maintain Golgi residence, and that COPI localizes predominantly to early Golgi compartments, so the finding that COPI inactivation does not prevent removal of Kex2 from the hybrid Golgi compartment is not unexpected. Further, I don't find the argument that Kex2 recycles within the Golgi (rather than from endosomes to the Golgi) to be sufficiently substantiated.*

We got ahead of ourselves in the initial submission by inadequately justifying the proposal that Kex2 recycles within the Golgi. That limitation has been overcome with the new Figure 10 and accompanying supplementary data.

The COPI results inspired us to consider the possibility that late Golgi membrane proteins undergo intra-Golgi recycling. That idea has received surprisingly little attention. While pursuing it, we obtained evidence that yeast cells lack an early endosome compartment that is distinct from the TGN. Although that story is still coming together, last year we published a crucial result by showing that compartments previously termed early endosomes, as judged by labeling with newly internalized FM 4-64, are actually identical to the Sec7-labeled TGN (M. Bhave et al., 2014, J. Cell Sci. 127:250-7).

If we provisionally accept that there is no yeast early endosome distinct from the TGN, then a simple argument can be made:

• AP-1 has been shown to mediate recycling of a subset of late Golgi membrane proteins in yeast;

• AP-1 has been shown to localize to the TGN. Our ongoing work confirms data from the Payne lab indicating that AP-1 is found on Sec7-labeled compartments;

• The implication is that AP-1 mediates recycling of a subset of late Golgi proteins from the TGN to earlier Golgi compartments.

In light of this working hypothesis and the data presented here, a reasonable supposition is that Kex2 undergoes AP-1-dependent and COPI-independent intra-Golgi recycling. We are excited about this unconventional model, which is shown in the new Figure 12.

*3) This work confirms, and modestly elaborates, that secretion of some proteins continues in COPI temperature sensitive mutants (Gaynor and Emr, 1998), which was shown many years ago (as the authors point out).*

The reason for our emphasis on the Gaynor and Emr (1997) paper is that it was the most careful prior analysis of the role of yeast COPI in Golgi traffic, and it was apparently inconsistent with cisternal maturation. At first we suspected that even the strong *sec21-3* mutation was unable to block COPI function completely. But after optimizing the anchor-away method, we concluded that yeast can indeed secrete without functional COPI. That result urgently needed an explanation.

Our data now provide a plausible explanation. When yeast COPI is inactivated, the Golgi forms a hybrid structure that can presumably continue to receive input from the ER, and late Golgi components of this structure continue to cycle in a manner consistent with ongoing secretory activity. We have therefore addressed the primary argument against an involvement of COPI in yeast Golgi maturation.

*Reviewer #2:*

*[…] The study is technically very thorough and the factual statements are convincing and well supported. We feel, however, that it has limitations, in that it leaves some essential question open and inconsistencies unresolved. Certain key issues should be addressed by suitable arguments and/ or by additional experiments.*

*The main problem relates to the difference between the previous results based on thermosensitive mutants and the present ones based on the anchoring away approach. The anchoring method, as noted by the authors, results in the adhesion of the whole Golgi cisternae to the mitochondria, rather than in the 'removal' of the COPI machinery from Golgi cisternae. The results of this are hard to predict and may not always be congruent with the specific inactivation of the concerned proteins. It is conceivable that the strong block in recycling of the cis-Golgi residents might be due to extensive proximity/adhesion between Golgi and mitochondrial membrane, rather than to a specific block of the COPI function. This possibility undermines the author's central conclusions about the role of COPI in the recycling of cis residents and should be appropriately discussed or addressed experimentally. Some relevant questions and considerations and some possible approaches are suggested below:*

*1) A re-examination of the ultrastructure of the Golgi-mitochondria contact due to anchoring might help. How extensive are the contacting surfaces? Does the morphology of these contacts justify the above doubts about the effects of anchoring on recycling?*

*2) Could the anchoring of the Golgi membranes to mitochondria through a COP-unrelated cis-Golgi protein reproduce the recycling block?*

*3) The authors obtain similar results by anchoring COPI to ribosomes, which might be less likely than mitochondria to severely obstruct membrane exchanges at the Golgi surface. Exploiting and characterizing the ribosome data at the ultrastructural level might help resolve the issue.*

*4) Another approach might be to consider that while early Golgi markers are blocked, late markers keep cycling. How are they disposed with respect to the contacting Golgi and mitochondrial surfaces?*

We were not overjoyed to learn that when COPI was anchored to mitochondria, the Golgi cisternae were anchored as well. This experiment arguably falls in the “one control too many” category. Yet the Golgi-mitochondria association is not surprising in hindsight because Golgi cisternae in *S. cerevisiae* are mobile, and nothing should prevent anchored COPI from interacting with and capturing mobile cisternae. Fortunately, all of our data, including the new Figure 4, indicate that mitochondrially anchored COPI is disabled for membrane traffic.

We believe that the concerns and suggestions from the reviewer are addressed by the ribosomal anchoring data. If mitochondrial anchoring had specific effects, then ribosomal anchoring should have different effects, but we saw essentially identical results with the two types of anchor.

A more extensive analysis of the Golgi-mitochondria association could be carried out. However, that approach is very labor-intensive, and we are unsure how much more insight would be gained.

We hope the reviewer will appreciate that our characterization of COPI anchoring to mitochondria is an attempt to be thorough in exploring the strengths and limitations of the anchor-away method. In the end, our judgment is that this method provides an efficient way to inactivate COPI.

*Reviewer #3: 1) One potential concern is that the anchored-away COPI proteins are still associated with the Golgi. Although cell growth is blocked by anchoring COPI, given the resolution of the growth curve data it is a formal possibility that partial COPI activity remains, and growth may not halt until after 1 or 2 more cell divisions. This might affect the interpretation of the data (especially the secretion phenotype), if partial COPI activity remains. The authors imply that because the sec21-3 mutant displays an identical secretion phenotype to the anchor-away COPI phenotype, it is likely that both methods result in complete inactivation of COPI. However, I think the authors need to demonstrate that the Golgi-ER COPI trafficking pathway is completely blocked by anchor-away inactivation of COPI. There are various "retrieval assays" in the literature. Another option would be to test by microscopy whether a protein that normally cycles between the ER and Golgi (i.e. a p24 protein or other cycling ER cargo receptor) is now exclusively found in the Golgi.*

We note that a gradual increase in OD_600_ is common for *sec* mutants at the nonpermissive temperature as the cells become denser, and a similar effect could explain the small increase in OD_600_ after anchoring COPI.

The reviewer points out that we cannot fully exclude the possibility that COPI retains residual activity. We have done our best by anchoring different COPI subunits alone or in combination, and by varying the rapamycin concentration and the duration of rapamycin treatment. No matter how hard we tried to inactivate COPI, secretion persisted. Those results fit with the data from Gaynor and Emr (1997), who used the strongest available COPI thermosensitive mutants. Ultimately, we concluded that yeast cells can almost certainly continue to secrete at some level in the absence of functional COPI.

Given the importance of this point, an additional control is merited. Thanks to the reviewer for the excellent suggestion of examining Golgi-to-ER retrograde traffic. The Sec71TMD-GFP construct from Sato and Nakano proved to be a suitable marker because it localizes to the ER by COPI-dependent recycling of the Rer1 retrieval receptor. In the new Figure 4, we show that anchoring COPI causes Sec71TMD-GFP to be depleted from the ER, as predicted if anchoring COPI blocks Golgi-to-ER recycling. This control provides a further validation of the anchor-away method.

*2) In the second half of the Discussion, the authors propose that AP-1 traffics Kex2 directly from late Golgi to early Golgi compartments. This would be a fairly radical idea, and would upend the current model that AP-1 traffics Kex2 from the Golgi to endosomes, and then Kex2 is retrieved from endosomes back to the Golgi by other machinery. Yet the only data provided specifically in support of this new model is the statement "our video microscopy revealed that most and perhaps all of the Kex2-labeled compartments acquire Sec7, suggesting that Kex2 recycles from older to younger cisternae within the Golgi". I disagree with both the data and interpretations contained in this statement. First, by watching their videos it appears that there are a few smaller Kex2 structures that may disappear without acquiring Sec7. Second, if we assume the first part of the statement is correct, this observation is consistent with Kex2 being retrieved from endosomes (where it may reside for only a few seconds), via vesicles, to Golgi compartments that have not yet acquired Sec7. To my knowledge, the precise Golgi compartment that fuses with endosomally-derived vesicles has not been firmly established in yeast, but the Nakano lab found that Ypt6 precedes Sec7 at the Golgi (PNAS 2013). As Ypt6 (together with the GARP complex) is primarily responsible for tethering/fusing vesicles from endosomes, this suggests Kex2 retrieval from endosomes to the Golgi occurs at a stage of Golgi maturation just prior to acquisition of Sec7. Therefore, it seems unfounded to propose a role for AP-1 in recycling cargos from late Golgi to early Golgi compartments.*

We are indeed proposing a fairly radical idea, and many in the field would likely have responded with similar skepticism to our initial submission. The revised manuscript tackles these issues with additional data and discussion. Our analysis began with the perception that current ideas about Kex2 trafficking are murky:

• One model is that Kex2 traffics in a similar manner as the vacuolar hydrolase receptor Vps10, by traveling to prevacuolar endosomes with the aid of Gga1/2 and then recycling to the Golgi. But this idea is puzzling because unlike Vps10, Kex2 has no obvious reason to visit prevacuolar endosomes;

• Another model is that Kex2 recycles from an early endosome to the Golgi with the aid of AP-1. But the molecular identity of this putative early endosome is unclear. Moreover, AP-1 localizes to the TGN.

Two results from our work support a simpler model in which Kex2 recycles within the Golgi. First, as shown in the new Figure 10, Vps10 is readily detected in prevacuolar endosomes but Kex2 is not, suggesting that these two proteins follow different itineraries. Second, our video microscopy data demonstrate that Kex2 arrives at the Golgi shortly before Sec7, and then departs shortly before Sec7.

Prompted by the reviewer’s comment, we carefully reexamined Video 6, which shows 4D microscopy of cells expressing Kex2-GFP and Sec7-mCherry. The results are presented in the new Video 7 and summarized in the text. We identified the structures that exclusively contained Kex2-GFP and that could be tracked for at least 30 sec. All of those Kex2-GFP-containing structures soon acquired Sec7-mCherry.

Our tests cannot rule out very transient passage of Kex2 through an endosomal compartment, and follow-up work will tackle this question with additional experiments. However, given the data presented here and the relative weakness of the published evidence supporting other views of Kex2 trafficking, we think it is justified to use Kex2 as a model late Golgi transmembrane protein for studying cisternal maturation.

If Kex2 and other late Golgi proteins recycle within the Golgi, candidate tethers include Ypt6 effectors. GARP functions in endosome-to-Golgi traffic but could also function in intra-Golgi recycling. Importantly, another Ypt6 effector is Sgm1, and its mammalian homolog is TMF, which was recently implicated in the intra-Golgi recycling of late Golgi proteins (Wong and Munro, 2014). These points are now emphasized in the Discussion.

[Editors' note: the author responses to the re-review follow.]

*The reviewers recommend that you address the following points with revisions to the text:*

*1) Reviewer 1 and reviewer 2 raise the issue that the anchor away approach that you employed results in different outcomes for Golgi residents when compared to the sec21-3 mutant and a Vps74/Golph3 mutant. They feel that this should be directly addressed in the Discussion. The comments from Reviewer 1 and 2 (below) elaborate this point.*

After prolonged treatment with rapamycin, GFP-Vrg4 shows progressive accumulation at the vacuolar membrane and plasma membrane, as would be expected given the block in recycling. This finding is now stated in the Results section as “data not shown”. Because there is actually no discrepancy between the expected and observed results, we chose not to elaborate in the Discussion.

*2) Since the arrested cisterna ('hybrid compartment') appears to remain competent to receive ER-derived material, and to export material, a brief discussion of the implications of this with regard to interpreting the results from both a cisternal maturation mechanism and a stable compartment mechanism of transport should be included in the Discussion. The comments from reviewer 2 are most relevant for this point.*

The last paragraph of the Discussion has been rewritten to address this issue more directly. We propose that Golgi compartmentation and COPI-driven cisternal maturation are aspects of the same phenomenon. Cisternal maturation is only needed, and can only occur, in the context of a compartmentalized Golgi. When Golgi compartmentation is lost, the yeast secretory pathway apparently operates in an alternative mode in which a single hybrid compartment receives COPII vesicles and produces secretory vesicles.

*3) Can the authors clarify in the text which of the studies cited in the Discussion (tenth paragraph) demonstrated that AP-1 was involved in Kex2 trafficking and describe the Kex2-localization phenotype(s) seen previously by others in AP-1 mutant strains?*

We had accidentally omitted the Abazeed and Fuller reference, which provides indirect evidence that AP-1 influences the localization of Kex2. That oversight has been corrected, and the discussion of AP-1 has been modified to emphasize that a role for AP-1 in Kex2 recycling is currently speculative.

*The entire reviews from each of the referees are included below to help you appreciate their suggestions. Addressing any other points raised in the reviews is left to your discretion.*

*Reviewer #1: In light of the observation that the anchor away approach results in entire COPI-coated cisternae to be associated with mitochondria, a major point of discussion amongst the reviewers during review of the earlier version of the manuscript regarded the question, does the technical approach (anchor away) impinge on cisterna function independently of COPI? It is puzzling to me that in the sec21-3 mutant (permissive temperature) GFP-Vrg4 transits the Golgi and is sorted to the vacuole, but it remains in place (in the hybrid compartment) in anchor away cells. The reason may lie in incomplete inactivation of COPI in the sec21-3 mutant (the figure shows permissive temperature), but difference in outcome is striking and begs resolution. The only way that I can think of that might meaningfully address this would be to compare results with 'inert' anchor away targets. What, for example, happens when if FRB is appended to a Golgi resident (e.g., Vrg4), so that the cisterna is addressed to the mitochondrion, but COPI is not manipulated? Despite this shortcoming, I think that in its present form the manuscript makes an important contribution to the field, even though the message, as summarized in the title, may be open to continuing debate.* As described above, we now state that GFP-Vrg4 accumulates in post-Golgi compartments after prolonged treatment with rapamycin.

*Reviewer #2:*

*In this study, the authors test the effects of inactivating the COPI complex on cisternal maturation and protein secretion in yeast. Previous studies with thermosensitive COPI mutants had indicated that COPI inactivation inhibits the secretion of many but not all proteins (Gaynor and Emr), and that it slows down, but does not block, cisternal maturation as visualized by live microscopy (Matsuura-Tokita and Nakano).*

*The authors attribute these inconclusive results to the fact that the available ts mutants can be either still partially active (Matsuura-Tokita and Nakano) or, when fully inactive, can markedly perturb the morphology of the Golgi (this manuscript), making it impossible to monitor directly the maturation process by live microscopy.*

*To resolve these uncertainties, they use the anchoring-away technique applied to specific COPI subunits. They find that COPI inactivation results in complete block of the recycling of the cis Golgi marker Vrg4, which now permanently labels the cis cisternae. This indicates that Vrg4 normally recycles through COPI. Moreover, they find that trans-Golgi residents (Sec7 and Kex2) partially merge with these stabilized cis cisternae, forming hybrid cis/trans Golgi bodies; and also that these trans markers cycle on and off the hybrid structures (though with somewhat delayed kinetics). This suggests that that* that *these markers cycle via a COPI-independent mechanism.*

*The study is technically very thorough and the factual statements are convincing and well supported. My comments will therefore concern only the novelty and the significance of the results.*

*As the authors emphasize, their data are the first direct demonstration that COPI is required for cisternal maturation. Evidence that COPI is required for maintaining the localization of Golgi residents (via Vps74/Golph3) has been available for some time (e.g., Tu and Banfield), but these previous observations are compatible both with the stable cisternae model and the cisternal maturation model. Thus, while not unexpected, these results are an important step towards establishing the molecular mechanism of cisternal maturation, and in principle deserve publication in a major journal such as eLife. They also raise the following questions and comments. The stabilization of cis Golgi markers in cis cisternae probably results from the specific type of COPI inactivation that is induced by the anchoring-away approach, which causes COPI to remain bound to the cis Golgi and so possibly to retain cis markers in the cis cisterna. However, the stabilization of the cis Golgi it is not the only possible outcome of COPI inactivation, under the maturation model. One might imagine, for instance, that if COPI were inactivated in a different way, e.g., by rapid degradation, this might cause cis markers to move forward along with cargo proteins to reach the vacuole or the plasma membrane. This is actually what happens when the COPI-bound adaptor Vps74 is removed: the Golgi enzymes leave the Golgi and move to the vacuole (Tu-Banfield). This apparent discrepancy might disconcert the reader. I therefore feel that the above point should be discussed, even though it does not in my view undermine the conclusion that COPI is needed for cis Golgi maturation.*

This point is important, but as described above, there is actually no discrepancy because anchoring COPI eventually leads to the appearance of GFP-Vrg4 in the vacuolar membrane and plasma membrane.

*The authors appear to interpret the data in the framework of the cisternal maturation scheme. However, the observation that the hybrid compartment can still absorb input from the ER and release secretory material towards the surface, albeit at slow rates, is difficult to reconcile with the maturation model. Since the recycling of cis markers is inhibited, and the formation of a new cis cisterna (a necessary step in the cisternal maturation mechanism) requires such recycling, a logical suggestion deriving from these data would be that the hybrid Golgi incorporates cargo and membrane from the ER by a mechanism different from cisternal maturation. In my view, these results are consistent with the idea that the main intra-Golgi transport mechanism is cisternal maturation, but they also suggest the existence of secondary mechanisms that are used by some protein classes, and/or of backup mechanisms used by the system when maturation fails. Again, I feel that, in order to avoid confusing the reader in a field that is already rich in controversies, the authors should discuss these issues, which, incidentally, are referred to also in the Introduction.*

The arguments here are somewhat subtle. As described above, we modified the Discussion in an attempt to clarify our ideas about how the Golgi operates in unperturbed cells and in cells with inactivated COPI.

*Reviewer #3:*

*This work convincingly clarifies the role of COPI, providing further evidence for a primary function in retrograde trafficking that is important for Golgi maturation. I think the authors now do a good job of explaining why some cargos are more sensitive to COPI inactivation than others, and how a basal level of secretion can be supported in the absence of COPI function. Furthermore, they have clarified the itinerary of Kex2. Therefore, I am in support of publication. My only remaining concern is a relatively minor one, which I would expect could be addressed through small changes in the text: Can the authors clarify in the text which of the studies cited in the Discussion on (tenth paragraph) demonstrated that AP-1 was involved in Kex2 trafficking and describe the Kex2-localization phenotype(s) seen previously by others in AP-1 mutant strains? In other words, what is the existing evidence in the literature that led to the model of AP-1 functioning in Kex2 traffic between the Golgi and endosomes? Furthermore, can the authors briefly discuss the established role of AP-1 in mammalian cells (i.e., papers from Bonifacino and other labs), and whether the authors think AP-1 has similar or different functions in yeast? I think explicitly addressing these points will help readers to better understand the insights from the current data in light of previous studies.*

As described above, we added the relevant reference about AP-1 and Kex2, and revised the accompanying text to emphasize that an involvement of AP-1 in Kex2 recycling needs further investigation. We also now briefly discuss how studies of mammalian AP-1 are potentially consistent with our model for AP-1-dependent recycling in yeast.